# Take a seed! Revealing Neolithic landscape and agricultural development in the Carpathian Basin through multivariate statistics and environmental modelling

Michael Kempf[1,2]*

**1** Department of Archaeology and Museology, Faculty of Arts, Masaryk University, Brno, Czech Republic,
**2** Physical Geography, Institute of Environmental Social Science and Geography, University of Freiburg, Freiburg, Germany

\* kempf@phil.muni.cz

## Abstract

The Carpathian Basin represents the cradle of human agricultural development during the Neolithic period, when large parts were transformed into 'cultural landscapes' by first farmers from the Balkans. It is assumed that an Early Neolithic subsistence economy established along the hydrologic systems and on Chernozem soil patches, which developed from loess deposits. However, recent results from soil chemistry and geoarchaeological analyses raised the hypothesis that extensive Chernozem coverage developed from increased land-use activity and that Early Neolithic 'cultural' groups were not restricted to loess-covered surfaces but rather preferred hydromorphic soils that formed in the floodplains. This article performs multivariable statistics from large datasets of Neolithic sites in Hungary and allows tracing Early to Late Neolithic site preferences from digital environmental data. Quantitative analyses reveal a strong preference for hydromorphic soils, a significant avoidance of loess-covered areas, and no preference for Chernozem soils throughout the Early Neolithic followed by a strong transformation of site preferences during the Late Neolithic period. These results align with socio-cultural developments, large-scale mobility patterns, and land-use and surface transformation, which shaped the Carpathian Basin and paved the way for the agricultural revolution across Europe.

## Introduction

Agricultural development in the Carpathian Basin played a major role in the transformation to an early domestic subsistence economy during the Neolithic period when it was transformed into a 'cultural landscape' by first farmers from the Balkans [1–4]. This stage became the starting point for the expansion towards the continent's northerly and westerly regions, which were formerly populated by scattered hunter-gatherer groups [5, 6]. The Carpathian Basin marked the northernmost boundary of the expansion of the Anatolian-Balkanic agricultural civilisation, as embodied by the Körös, Criş and Starčevo 'cultural complexes' in the first half

**Data Availability Statement:** The data underlying this study contain information that can accelerate the destruction of cultural heritage through looting and cannot be shared publicly. The data presented

in this article are available upon request from Dr. Attila Kreiter, Hungarian National Museum at Budapest, contact information: archeodatabase@hnm.hu or directly from the Hungarian National Museum Institutional Data Access / Ethics Committee (contact via https://mnm.hu/en) for researchers who meet the criteria for access to confidential data.

**Funding:** Michael Kempf received funding from the Operational Programme Research, Development and Education - Project „Postdoc2MUNI" (No. CZ.02.2.69/0.0/0.0/18_053/0016952) The funders had no role in study design, data collection and analysis, decision to publish, or preparation of the manuscript.

**Competing interests:** The authors have declared that no competing interests exist.

of the 6th millennium cal BC [7–10]. The flat eastern part of the Alföld region faced a development more intensely tight to the northern Balkans than to Central Europe. The western part was a scene of swift migrations and renewed impacts from the northern Balkans. Transdanubia belonged to the south-eastern periphery of the so-called Linearbandkeramik 'culture' (LBK), with consensus that this was the specific area of its origin [11]. At the onset of the 5th millennium cal BC, 'cultural' and genetic impulses arrived again from the southeast towards Transdanubia that helped to shape the large Lengyel 'cultural' formation in Central Europe [11, 12]. Migration and mobility played a key role in the Neolithic of the Carpathian Basin, stronger effecting its western part [4, 9, 13–18].

So far, an early domestic subsistence economy and a mosaic of permanent settlements and transient camps on elevated loess-covered areas with thick and fertile Chernozem soils in close connection to fresh water has been assumed [19–23]. However, recent results from site location modelling and soil organic matter analysis raised the hypothesis that Neolithic and subsequent land-use activity strengthened the development of extensive Chernozem coverage through intensified Black Carbon (BC) input from natural wildfires and human-induced periodical vegetation burning [24–31]. Extensive Chernozem soil coverage on Pleistocene loess deposits would be a result of intensive surface transformation rather than a prerequisite of Early Neolithic agricultural developments. Building up on previous results [26], this article presents site preference analysis of Early Neolithic to Late Neolithic site locations in Hungary based on spatial data provided by the Hungarian National Museum and the catalogue edited by Anders and Siklósi (2012) (**Fig 1**). The large dataset with 677 Early Neolithic sites from the so-called Körös 'culture' and the Hungarian database with 2662 sites spanning the entire Neolithic allows for a comprehensive and statistically profound site location model and the evaluation of land-use and agricultural strategies, environmental dynamics, and feedbacks as well as landscape development during the Neolithic period in the core area of the European agricultural revolution.

## Environmental settings

Hungary is dominated by a moderate climate with marine and Maritime influences and increasing continentality towards the central plain [32–34]. Harsh Early Holocene climatic conditions limited tree growth, increased sand accumulation [35, 36], and led to the development of a forest-steppe vegetation followed by a subsequent forest decline and niche habitat survival [35, 37–40] until warmer conditions initiated the Atlantic phase [21, 41, 42]. Large parts of the Great Hungarian Plain are covered with Quaternary sediments (**Fig 2**) [43] and Upper Pleistocene loess is characteristic for the hilly margins of the plain, the Mezőföld west of the Danube, and the alluvial fans of the basins [44]. The loess alternates with sandy layers and palaeosoils, which developed during warmer and wetter interglacial periods [45]. The eastern Carpathian Basin is characterized by Holocene hydrologic floodplain dynamics of the river Tisza [46–49], the river Körös [50], and the river Maros [51], while the western part is dominated by channel outbreaks of the anastomosing river Danube.

Nine different soil types can be distinguished in Hungary, from which meadow soils (*Humic and Mollic Gleysols; Gleyic Phaenosems and Chernozems*), brown forest soils (*Cambisols; Luvisols; Umbrisols*), and Chernozems (*Chernozem; Phaeozems; Kastenozems; Vertisols*) are mostly abundant [52, 53] (**Fig 2C**). Sandy soils (*Arenosols; Cambisols*) and salt affected soils (*Solonchaks; Solonetz*) are locally clustered in the central DTI (Danube-Tisza Interfluve) and in the river Tisza floodplain region [26, 54–56]. Stony and lithomorphic soils are connected to the northern and western parts and the margins of the Carpathian Basin [57].

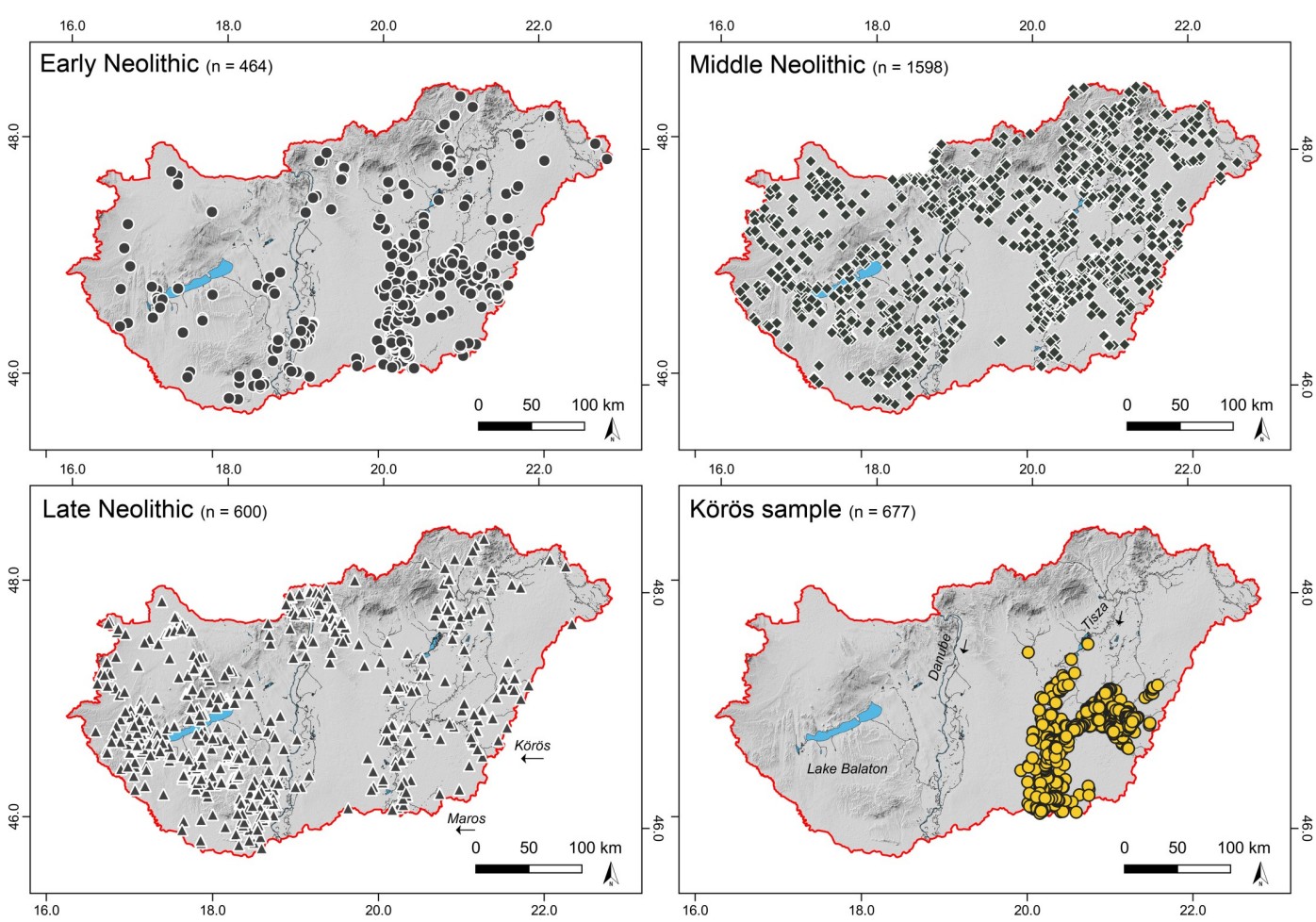

**Fig 1. Distribution of the Early (EN), Middle (MN), and Late Neolithic (LN) sites in Hungary.** The Early Neolithic Körös sample is located along the rivers Tisza, Körös, and Maros in the eastern part of modern Hungary (EN, MN, and LN sites provided by Attila Kreiter and the Hungarian National Museum at Budapest; Körös sample based on Anders and Siklósi, 2012).

The loess-covered plains are mostly dominated by Chernozems (*Pachic* and *Typic Claciustolls*) with distinct hydromorphic conditions in depressions (*Aquic Calciustolls* and *Haplustolls*) [58]. During dry glacial periods with low precipitation rates, salt affected soils developed [59] over saline groundwater outbreaks under increased evaporation [60, 61], particularly in the river Tisza floodplain [54, 61]. The strong Holocene environmental dynamics created a very heterogeneous soil mosaic and a highly diversified agricultural potential [26, 52, 53, 57, 62–64].

## Administrative settings and heritage regulations

Hungary is divided into 19 counties (*Komitaten*) plus Budapest. Each county has a County Museum and several town or city museums where rescue excavations are carried out and the finds are kept in magazines. Above these, there is the *Hungarian National Museum*, the *ELTE University*, and further Universities that have the right to dig anywhere and partake in the post-excavation work and publishing process [69]. Documents and finds were digitally stored in a central online database at the National Museum at Budapest (https://archeodatabase.hnm. hu/en, last accessed 30th of June 2021, contact: Dr. Attila Kreiter, archeodatabase@hnm.hu)

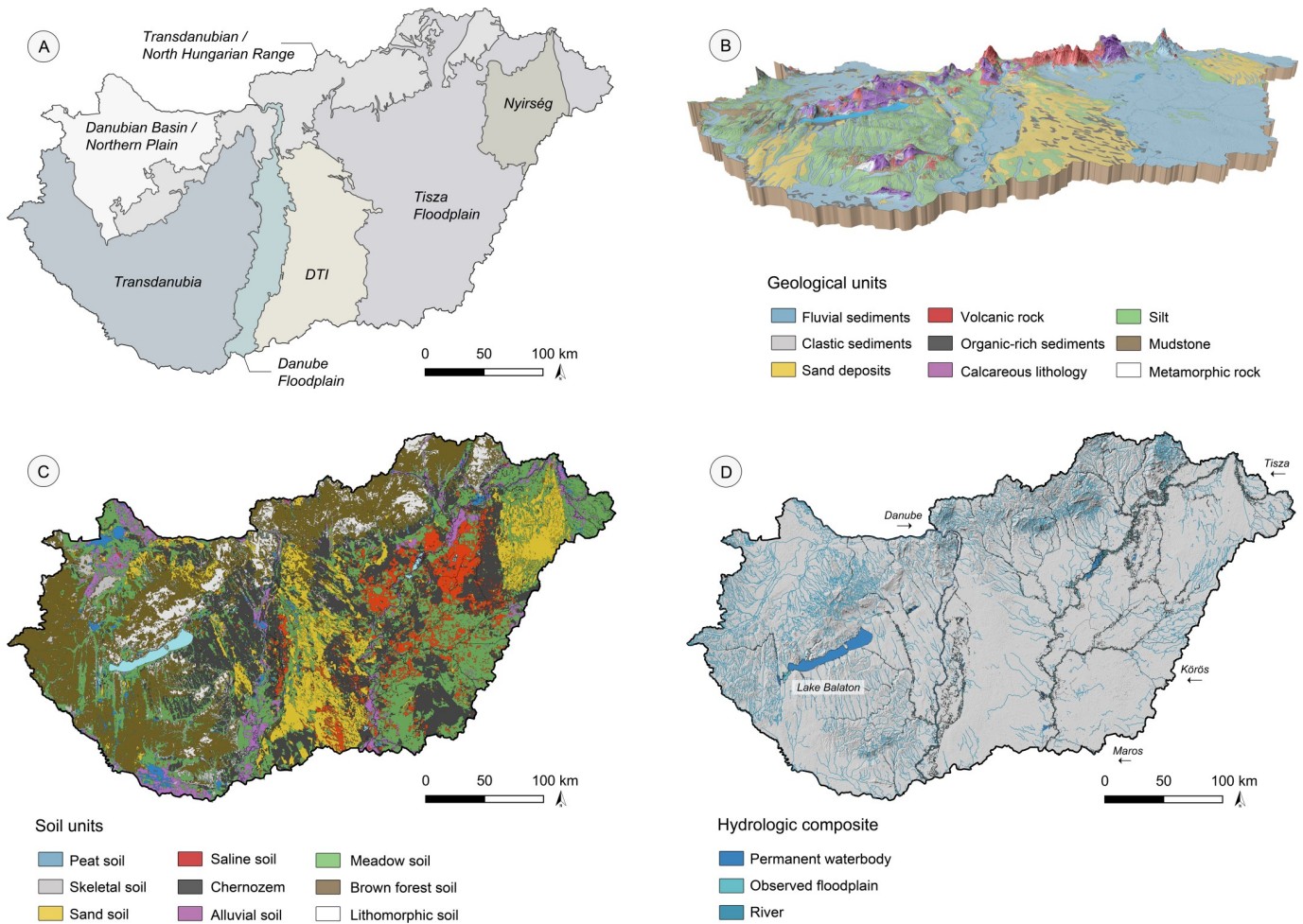

**Fig 2. Environmental settings of modern Hungary.** (A) Regional classification based on geological, hydrological, and pedological settings [16]; (B) 3-D model of the geological and topographical parameters (30-times vertical exaggeration) [65, 66]; (C) soil units in Hungary [57, 67]. For the translation to the WRB system see [26, 52, 53]; (D) hydrologic composite based on the observed floodplain extent [68], permanent waterbodies, and the modern river system.

[70, 71]. Furthermore, the official site register of the National Office of Cultural Heritage (KÖH) contains topographical data collected by the Archaeological Topography of Hungary Program (MRT).

For the construct and rescue excavation archaeology, prior to building motorways and further large-scale developments, the fieldwork (pre-excavation diagnostics, field surveys, LiDAR scan analysis, drone prospection, coring, etc.), the excavation, and the primary post-excavation documentation changed the institutions in-charge over the last 30 years at least six times. Political, socio-economic and administrative development in Hungary further kick-started large construction works and extensive surface transformation during the past 30 years [72]. Today, there is one state company that controls the entire country, with a lab that has protocols, for example regarding geological, palynological, and zooarchaeological data managed by the National Museum. But this is–along with many other things–still on the move and unfortunately, the triggering effect is not primarily the heritage protection but rather to facilitate building and construction activity by investors and 'land developers' [70], which have grown exponentially since 1990 [73].

Between 2007–2020 the dynamics are similar, amplified by an increase in motorways, car parks, shopping malls, and others–each of these needed a previous archaeological monitoring and excavation [72]. In the last few years, the legislation has become extremely restrictive, with only up to 30 days for any excavation [72, 74]–all this reflecting a picture that archaeology is seen as a main and hostile obstacle for any development, and not as a main source of cultural heritage.

Like in other regions [75–78], the regional and local extensive building activity described above and the concentration of archaeological sites along the construction corridors and among densely populated agglomerations impacts the distribution of archaeological features across Hungary. Consequently, the 'archaeological landscape' is not a realistic representation of the actual human-environment interaction sphere during the Neolithic period, but rather a simplified model of current land-use opportunities and landcover change. However, and as pointed out by Bánffy and Raczky (2010), there is a very high density of archaeological sites to be expected within the boundaries of the modern Hungarian state. The actual number of sites has been estimated to be at least 200.000 sites, which makes an average density of over two sites per km$^2$ [70]. Taken into account that parts of the Carpathian Basin are less suitable for agriculture than others (e.g., the Danube-Tisza Interfluve, DTI, see **Fig 2**) and consequently show a less dense archaeological coverage, it is a question, to which extent the site distribution is a function of the lower soil quality and water availability of the region or rather a modern bias of current agricultural and settlement probabilities–and hence mirroring less recent urban and infrastructural development (**Fig 3**).

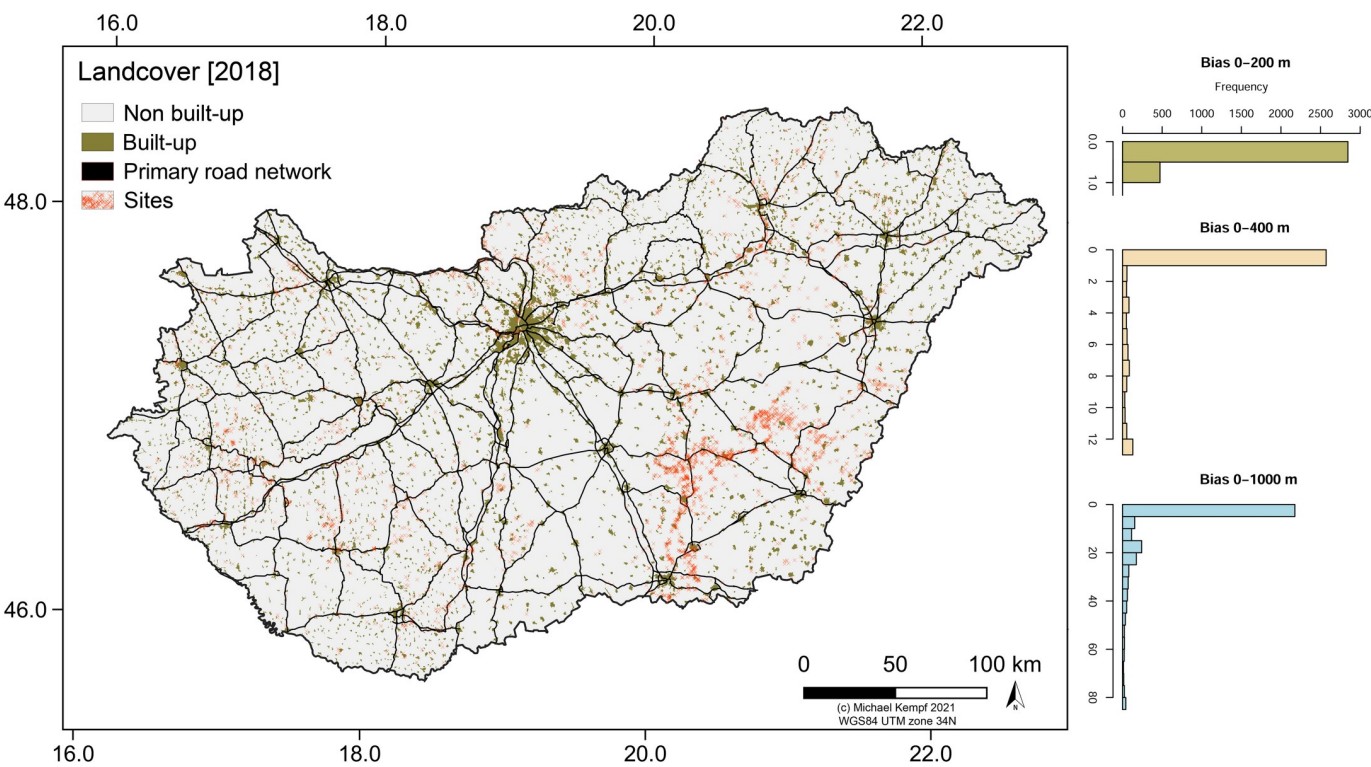

**Fig 3. Infrastructural development in Hungary.** Built-up and main traffic arteries (highways and 4-lane national roads) in 2018 and the frequency of archaeological sites within 100 (0–200 m), 200 (0–400 m), and 500 m (0–1000 m) radius around the bias surface.

## Material and methods

Archaeological data covering the Neolithic period was provided by the database of the Hungarian National Museum (https://archeodatabase.hnm.hu/, last accessed 26[th] of Nov. 2020). A comparison Early Neolithic Körös data sample was extracted from the catalogue by Anders and Siklósi [9]. The sites were grouped as Early (EN), Middle (MN), and Late Neolithic (LN) and spatially analyzed using open-source GIS (QGIS Geographic Information System. QGIS Association. http://www.qgis.org, last accessed March 2021) and R software (R: A language and environment for statistical computing. R Foundation for Statistical Computing, Vienna, Austria. URL https://www.R-project.org/, last accessed March 2021). In a first step, the sites were analyzed for their spatial behavior and Complete Spatial Randomness (CSR) using Point Pattern Analysis (PPA) such as Intensity or Kernel Density Estimates (KDE) and simulation envelopes such as the inhomogeneous G-Function and Ripley's K-Function [79–84] (**Figs 4 and 5**). The null hypothesis (*H0*) is that the data point pattern is a realization of complete spatial randomness. Intensity analysis [85] (expected number of points per unit area [86]) allows to track changing frequencies of observations in the data over a geographic extent. The interpolation (KDE) provides a smoothed visualization of the point pattern at different radii, through which the density levels were processed [76, 81, 86–88]. Among other possibilities, the radius (or bandwidth) of the KDE can be determined from CSR tests, which test for regular, clustered, or random point distribution [82, 84]. In this context, a broad variety of methodical approaches for PPA have been developed, which determine, whether an observed distribution of points is taken from a random point distribution or whether it forms clusters or follows regular spatial behavior [89]. Bailey and Gatrell (1995) pointed out that PPA can be divided into *first-order effects* and *second-order effects*, which refer to global variation in the mean value of the process and local deviation of the process from its mean value [85, 89, 90]. This means that first-order characteristics describe the average point intensity in a specific area and how this intensity varies due to external processes [85]. Consequently, these effects characterize the random, clustered, or regular point patterns. The second-order characteristics determine, whether the point distribution is affected by the spatial configuration of other points [85].

## Complete spatial randomness

Among others, Ripley's inhomogeneous K-Function is one of the most useful statistical tools to test for CSR [80, 81, 91]. According to Marcon and colleagues (2013), an observed set of points is tested against a homogeneous Poisson point process taken as a null model [92–94]. The function can further be used to determine the radius, at which clustered behavior can be detected [95] and follows the equation described in Nakoinz and Knitter (2016, 138) (all site distributions were calculated with 2000 random point simulations) [83, 84, 91, 93], in which S are events in C (circle with radius d and the events in the circle), divided by the overall study area event density $n\lambda$ ($n$ = number of all points and the $\lambda$ = intensity of the process):

$$K(d) = \frac{\sum_{i=1}^{n} \#(S \in C(s_i, d))}{n\lambda}$$

In general, the K-Function counts the number of points within given distances around each point and compares the result to the number of points one would expect within a totally random point distribution. If the number of empirically observed points within a certain distance is greater than the number of the simulated random distribution, the empirical point pattern is clustered at that scale. If the number is smaller than the simulation, the distribution is dispersed [95].

The G-Function cumulates the frequencies of nearest-neighbour, which is based on the higher likelihood to have close-by located points in clustered point patterns then in random

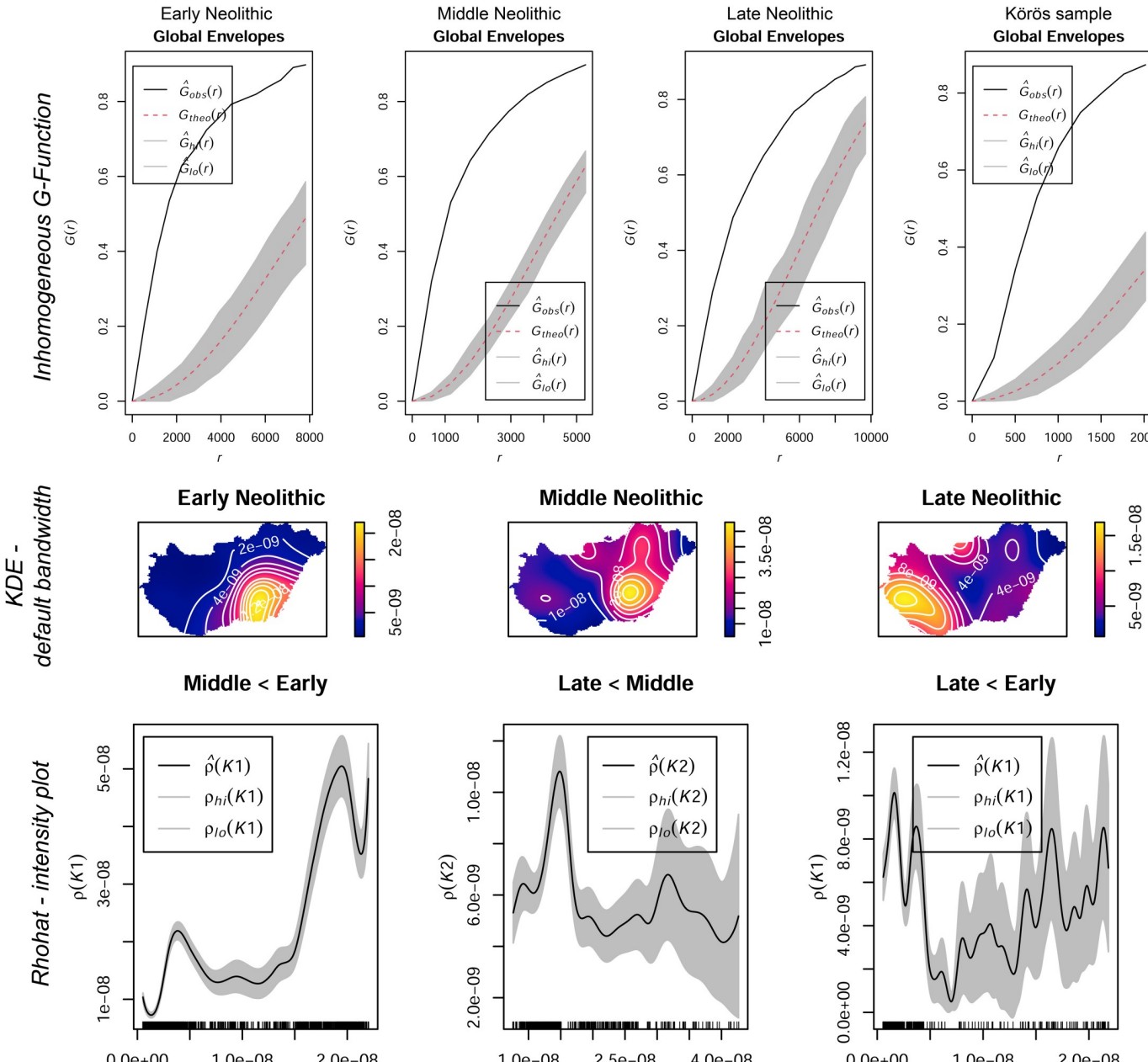

**Fig 4. Point pattern analysis.** Upper part: Envelope simulation of the G-Function for Early, Middle, and Late Neolithic sites as well as the Körös comparison sample. All observed samples show significant clustering above the envelope. Middle part: default bandwidth kernel density estimation of the sites. Lower part: *Rhohat* intensity plot using spatial point pattern and site distribution density as explanatory covariate. The function measures whether the Middle Neolithic site distribution is a function of the previous Early Neolithic site distribution (LN/MN and LN/EN respectively).

point distribution [84]. In a Monte Carlo simulation, envelopes were calculated for random points and a number of point patterns were produced to calculate the *G(d)*-values using the formula described in Nakoinz and Knitter (2016, 136):

$$G(d) = \frac{\{d_{min}(s_i) \le d\}}{n}$$

which calculates the fraction of all nearest neighbor distances $d_{min}(s_i)$ in the pattern that is less

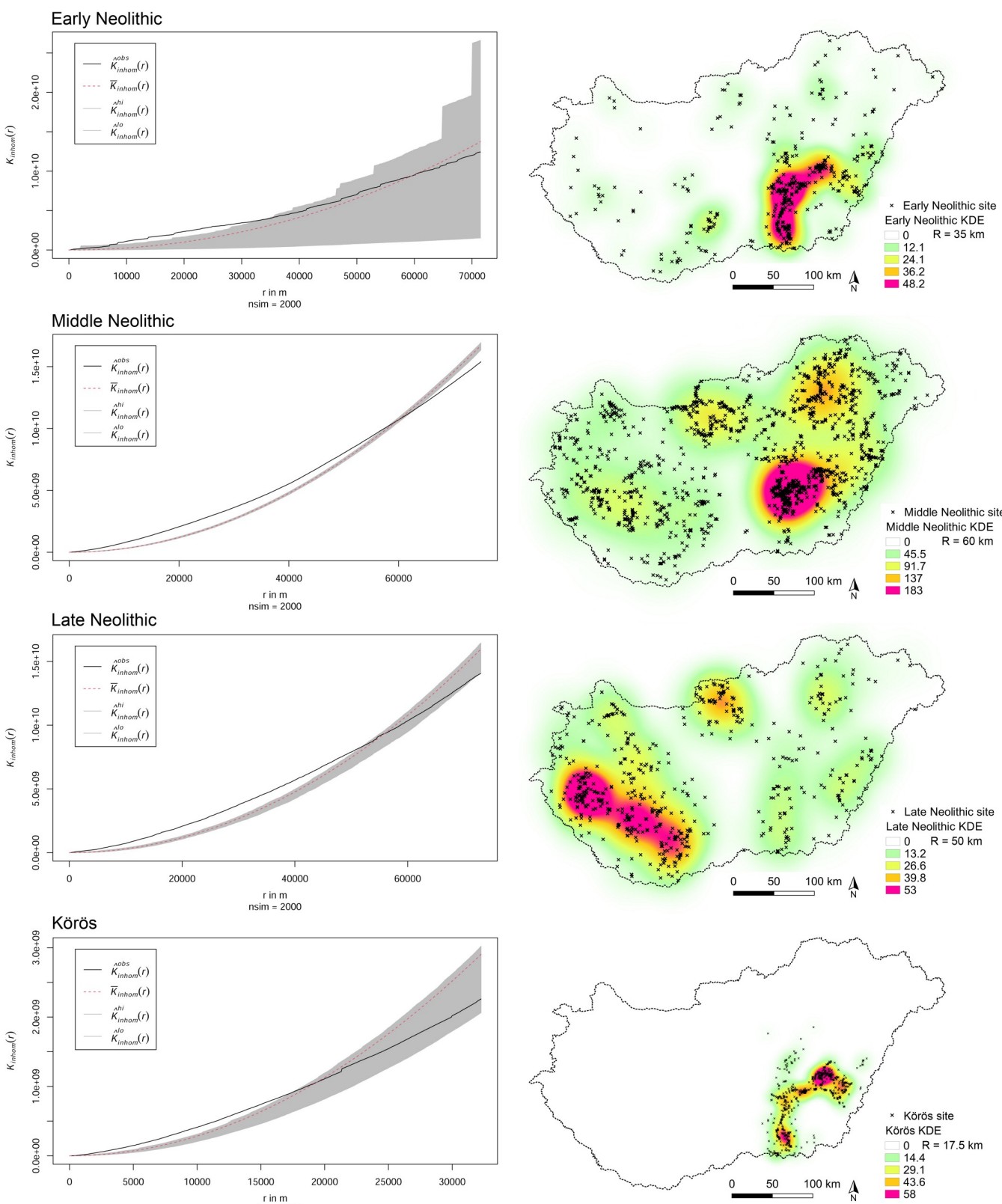

**Fig 5. Ripley's K-Function and Kernel Density Estimates (KDE) for early, middle and late Neolithic, and the körös comparison dataset at different bandwidths.** The KDE shows regional site continuity in the river Tisza region from the Early to the Middle Neolithic and a strong spatial shift towards Transdanubia during the Late Neolithic period.

than *d* divided by the number of events *n* [84].A Monte Carlo test has been performed using the r-package *spatstat* [96] and the *envelope()*-function [97]. This test is based on simulations from the null hypothesis. To calculate the envelopes, point pattern data (*Y* = Neolithic sites), a function that calculates the desired summary statistics for a point pattern (*fun* = Gest), the number of simulated point patterns to be generated when calculating the envelopes (*nsim* = 2000), and the envelope value rank (*nrank* = 1) are provided to the function [98]. A rank of 1 means that the minimum and maximum simulated values are used [84, 98]. The width of the envelope reflects the variability of the process under the null hypothesis of CSR [98]. The G-function was performed from this R code:

```
s <- readOGR("D:/data ","my_sites")
data <- as(s, "ppp")
data_smo <- density.ppp(data)
Window(data) <- Window.im(data_smo)
E <- envelope(Y = data, fun = Gest, nsim = 2000, nrank = 1)
plot(E, main = "Global Envelopes")
```

and the K-Function was carried out using the R code:

```
s <- readOGR("D:/data ","my_sites")
data <- as(s, "ppp")
data_smo <- density.ppp(data)
Window(data) <- Window.im(data_smo)
K_data <- envelope(data, fun = Kinhom, nsim = 2000,
     simulate = expression(rpoispp(data_smo)),
     correction = "best")
plot (K_data, main = "Kinhom Set 1", sub = "nsim = 2000", xlab = "r in
m")
```

## Point pattern analysis using spatial covariates

To measure the dependency of the point distribution to environmental parameters, the environmental raster data were tested separately against the spatial distribution of the point patterns. The R-package *spatstat* [96] was used to identify point pattern intensity in relation to a spatial covariate, which is a representative of a pixel image, e.g., a two-dimensional raster with categorical values [97]. First, a so-called window was created from the spatial extent of the point pattern to identify the area where points were observed and where points were not observed. The covariate serves as "explanatory variable" [97] and in this case, soil units, elevation, slope gradient, and hydrologic system were considered to be explanatory for the point pattern distribution. The original classification of the soil units is described in categorical values, which means that each soil class is assigned a discrete value (e.g., Chernozem = "50"). However, *spatstat* is operating with continuous covariates and the application of the *rhohat* function, which is desired in the approach of this paper, with discrete (classified) raster values (representing the covariates) is inappropriate [99]. To overcome this problem and to simultaneously consider not only the tight soil values of the respective soil distribution but also take into account the distance between points of the point pattern and the respective soil units, the soil data was reclassified into continuous values. From the soil raster, scale-based focal statistics were calculated from each raster layer with r = 1–5 km and r = 10 km distance around each cell using the *r.neighbors* function in GRASS GIS (GRASS Development Team, 2020. Geographic Resources Analysis Support System (GRASS) Software, Version 7.8. Open Source Geospatial Foundation. https://grass.osgeo.org, last accessed March 2021). This function refers to each cell of the raster layer and considers the surrounding cell values assigned to this cell in a user-defined neighborhood. Basically, each cell is given the summed-up values of the cells around it. A quick example helps to clarify the procedure: a pixel of value 1, which is enclosed by 8 pixels with equal values, will be assigned the value 8 if the neighborhood is set to 1 pixel.

The pixel on top will be assigned the value 6 and to the edges, the values decrease to 4. Consequently, depending on the radius, the center of each soil unit is assigned high values, which decrease towards the margins in a kernel-like value range. A two-sample *Kolmogorov-Smirnov-test* (KS-test) [84, 100] was performed. However, both tests take into account the 0-dimensional spatial variation of the point distribution and the underlying environmental variable without considering the catchment composition. Consequently, the *Vargha-Delaney A statistics* (VD-A) were performed to integrate variable conditions in the catchments of each site. VD-A statistics were performed from each focal raster (soil, geology, slope, elevation, water, and impact by modern built-up) with radii between 1 and 5 km. With the *point.sampling.tool*, all sites were assigned the values of the aggregated cells. Random point distributions, which equal the number of observed sites were used as comparison datasets for each period and each radius. For this reason, random points were simulated within the boundaries of the study area. The VD-A test compares the empirical to the random simulation and tests if they were drawn from the same distribution or if there are significant differences [101]. The EN Körös sample was further delimited by its own distribution area, which was buffered with 5 km. The KS-test was carried out following the statistics described e.g. by Nakoinz and Knitter (2016, 134; 7.2) [84, 100]:

$$D_{n,m} = \frac{sup}{x} |F_{1,n}(x) - F_{2,m}(x)|$$

In which $F_{1,n}(x)$ is the empirical cumulative distribution function and $F_{2,m}(x)$ the theoretical or the empirical [84]. The VD-A statistics were calculated using the formula:

$$\mathbb{A}_{12} = [E(R_1/m) - (m+1)/2]/n$$

first described by Vargha and Delaney (2000, 110) [101], where $E(R_1/m)$ is the expected value of the rank mean of sample 1 [101].

Calculations were done using R software and the package *Effsize* (Efficient Effect Size Computation) [102] using the following R code:

```
# Define the sites (= my_data) and a random comparison dataset, which
contain covariate information ## (soil raster value from focal
statistics)
    data <- read.csv("my_comparison_data.csv", header = TRUE)
    data2 <- read.csv("my_data.csv", header = TRUE)
# Perform the KS-test and VD-A statistics from the package Effsize for
the point pattern against a
## random point pattern
    ks.test(data2$my_soil, data$my_soil)
    VD.A(data2$my_soil, data$my_soil)
```

In addition to the VD-A statistics and the KS-test, the function *rhohat* in *spatstat* was used to estimate the dependence of the point intensity on a covariate [94, 96, 103]. In this case, the soil focal raster data at a radius of 10 km, the slope gradient and elevation (both represent continuous value ranges), and the 5 km focal focal statistic of the hydrologic system was used. The geological data was considered to be too generalized, particularly in the extensive floodplain of the river Tisza. The *rhohat* function produces a plot, which is an estimate of the intensity (*z*) as a function of each covariate. To read the shapefile, the *sf* R-package designed by Edzer Pebesma has been used [104]. The plot was generated following this code:

```
# load a window
    s <- st_read("my_window.shp")
    w <- as.owin(s)
# read the point shapefile
```

```
      s <- st_read("my_data.shp")
      data <- as.ppp(s)
# load the soil raster
      img <- raster("my_soil.tif")
      soil <- as.im(img)
# Compute rho using the ratio method
      rho <- rhohat(data, soil, method = "ratio")
# Generate rho vs covariate plot
      plot(rho)
```

For the interpretation of the *rhohat* plots, it is helpful to estimate the point density in comparison to the respective covariate because the plot describes the intensity of observations/the point pattern as a function of the covariate data. For example, the re-classified soil raster can be seen as a density estimation of the soil with the highest values in the center of the actual distribution and lower values towards the margins and within a 10 km distance around the soil patches. A point intensity analysis can easily be performed using a kernel density estimation in *spatstat* following the code based on [94]:

```
s <- st_read("my_window.shp")
w <- as.owin(s)
s <- st_read("my_data.shp")
data <- as.ppp(s)
Window(data) <- w
img <- raster("my_soil.tif")
soil <- as.im(img)
Window(soil) <- w
K1 <- density(data) # Using the default bandwidth
plot(soil, main = "my_soil-my_data")
contour(K1, add = TRUE, col = "white")
```

Site distributions were compared to environmental variables including digital soil data [57, 67], river systems and observed flooding extent including palaeochannels [68, 105], geological data [65], and a digital elevation model (AsterGDEM, resampled to 100 m resolution) [66]. KS-test and VD-A statistics were performed on the slope gradients and the elevation data to estimate the terrain roughness in the catchments and the preferences for flat, gentle slopes, or steep slopes and the overall elevation. The distance between the sites and the nearest available water access was modelled using the *distance.to.nearest.hub* tool in QGIS. Availability of water patches was estimated using VD-A statistics from a composition of the current river system and the Corine riparian delineations, which describe the observed extent of the floodplain (CLC, 2018). The distance matrix was plotted with the *ggplot2* package in R [106] and the VD-A statistics were performed from the resampled hydrologic composites with a grid size of 100 m. The geological data was classified into the criteria volcanic, silt, organic rich sediments, mudstone, metamorphic sediments, Holocene sand deposits, Quaternary fluvial deposits (diamicton), clastic sediments, and calcareous lithology. From the variables, the p-values of the VD-A statistics were plotted using a multiple line plot in *ggplot2*. All values > 0.5 suggest a rejection of the null hypothesis and indicate clustered behavior of points over the respective soil class. Values < 0.5 suggest avoidance of the soil class and no clustered behavior. The site distributions were cross-checked with the modern rural and urban built-up, residential areas, and infrastructure to distinguish the bias in the empirical model and evaluate site continuity [75].

To test the influence by modern built-up and infrastructural development in Hungary (see **Fig 3**), a bias comparison surface has been produced using current (2018) Corine Landcover data [68] and the primary road network [107]. First, Open Street Map (OSM) data was downloaded [107] and residential and industrial areas were resampled to a 100 m pixel size. The primary infrastructural arteries were buffered with a 100 m buffer and rasterized to 100 m

resolution. Both datasets, the infrastructure and the built-up, were merged to create a binary raster surface and a vector mask (1 = artificial surface, 0 = non-artificial surface). VD-A statistics and KS-test were performed from the raster data and a nearest-point distance matrix for each period was processed from the vector data. Furthermore, all archaeological sites were analyzed for their spatial behavior compared to the artificial built-up. First, histograms of the site distributions were generated, which visualize the differentiation of sites within and outside the binary artificial raster surface (see **Fig 3**). Using the *r.neighbors* function with a radius of 200 and 500 m, two raster surfaces were created, which take into account the potential impact and findability of sites by building activities within a specific 'construction buffer'. The kernel value, which was produced by the function was added to each site attribute table and visualized using histograms. The values range from high (center of a construction site) to low (margin of a construction site) and zero (distance greater 707 m (5 times the diagonal of a 100x100 m pixel). From the frequency of the distribution, the relationships between the sites and the current built-up were estimated.

Because Hungarian archaeology is basically controlled by rescue excavations, two point samples were further created from the total site dataset. The first covers all sites, which are directly located within the current built-up and infrastructure mask (n = 493) and the latter represents all sites outside the mask (n = 2461). Finally, a distance matrix was generated, which measures the distance between the site and the nearest built-up and/or infrastructural construction. The distance relationships were plotted to visualize the areas, which show increased (linear) site densities as a function of recent building activities.

## Results

From the PPA and intensity estimates (KDE), the clustered distribution patterns can be observed for all point samples. The G-Function examines the cumulative frequency distribution of the nearest neighbour distances and increases rapidly at short distances. The envelope simulation clearly shows that the observed point distribution is located above the envelope (**Fig 4**), which means clustered point patterns at any location. The same results derive from the K-Function, which also can be used to determine the radius at which clustered behaviour is established. If the observed black line lies above the envelope, the point patterns are clustered. The moment the line crosses the envelope, CSR is established: The envelopes were used to evaluate the goodness-of-fit of a point process model to point pattern data [94, 96]. The test rejects the null hypothesis (the data point pattern is a realization of complete spatial randomness) if the graph of the observed function lies outside the envelope at any value of *r*. The width of the envelopes show the variability of the process under the null hypothesis of CSR [98]. In this case, all point patterns show clustered behaviour at small scales and CSR at larger radii. According to the results, the KDE were plotted in a heatmap to visualize intensity estimates and site continuities over the chronological intervals (**Fig 5**).

The results from the KS-test indicate, whether the empirical sample and a random comparison dataset are drawn from the same distribution. The p-values show if this null hypothesis can be rejected or not. The results from the distance analyses show that peat soils but also Chernozem and more skeletal, sandy, and lithomorphic soils were close to the random comparison data. Hydromorphic soils like meadow and alluvial soils but also saline soil patches and brown forest soils show significant differences during the EN period (see **S1 and S3 Tables**). KS-test results from topographical variables confirm the significant differences between the random point distribution and the EN sample (see **S2 Table**). The same can be observed from the hydrologic composite data sample. Even though the geological data resolution is quite coarse, the KS p-values fit the results from the soil analysis and the hydrologic and

topographic sample. Quaternary fluvial sediments show very small p-values as well as sand deposits. Clastic sediments and metamorphic rocks, waterlogged mudstone and organic, peaty units also demonstrate significantly low values. It is remarkable that the Chernozem p-value seems to be reflected in the low values for silt, which is mostly referring to loess deposits in the study area. During the MN period, Chernozem became significantly different from the random distribution. This changed again during the LN period where brown forest soils and skeletal, stony soils became more significant. The results of the KS-test (see **S1–S3 Tables**) can confirm the differences of the sample but cannot show, which these differences are. The VD-A statistics and the point intensity and covariate function *rhohat* provide more powerful tools to take into account the environmental conditions within the catchment of each site.

## Soil preferences

The Hungarian soil characteristics can be considered very heterogeneous and mosaic-like in most of the parts. Consequently, the country has been divided into seven major regions, which were classified from geological and pedological data (**Fig 2A**) [17]. These regions show distinct environmental settings and geographical characters based on the hydrologic system, sedimentology, and topographic features, which can be used to detect distribution patterns and the spread of the 'cultural' expansion throughout the Neolithic. Building on the regional diversity, the EN site locations are clustered in the south-eastern part of Hungary with a significant concentration along the hydrologic system of the rivers Tisza, Maros, and Körös. In this geographic range, landscape features can be described as topographically flat surfaces characterized by Quaternary fluvial deposits and hydromorphic meadow and alluvial soils interspersed with silty deposits from aeolian dust accumulation, which produced patchy loess-covered islands on palaeolevees standing out from the floodplain sediments (**Fig 2B**).

## Soil preferences during the early Neolithic

A strong preference for hydromorphic meadow soils and a medium preference for alluvial soils can be detected during the EN, which is displayed by the *p-values* of the VD-A statistics (**Figs 6 and 7**). From the *rhohat* plots, a more precise conclusion can be drawn to which extent the site catchments were dependent on environmental covariates. The soil data shows clear preference for alluvial soil and meadow soils with a maximum of site intensity at highest values–indicating the core of the kernel and hence the direct location within the respective soil patch. During the EN, brown forest soils as well as skeletal, sand, lithomorphic, and peat soils did not play a decisive role in the catchment compositions of the sites (**Fig 8A and 8B**). The *rhohat* plots of these soil covariates show only scattered site distributions or no sites at all in a distance of up to 10 km around the sites.

There is also a very strong correlation between site location and availability of salt affected soils within close distance to the site. The *rhohat* plots further strengthen this argument with increased intensity of sites at lower and medium covariate values. This points towards the importance of salt patches in the close vicinity but not immediately at the site. The second peak at maximum values, indicating direct location parameters can be considered spatial outliers. These saline soils are not favorable for crop cultivation and indicate rather poor drainage capacity, periodical waterlogged conditions, and high evapotranspiration that exceeds the average precipitation rates. However, salt exploitation played a major role during the neolithization process with increased livestock breading and the shift from a salt- and protein-rich meat-based diet towards a cereal-based diet with a low salt-content [108, 109]. The correlations are not only geographically controlled by the regional distribution of the respective soil units. A comparison dataset of 677 sites occupied by the EN Körös 'culture', which was calculated

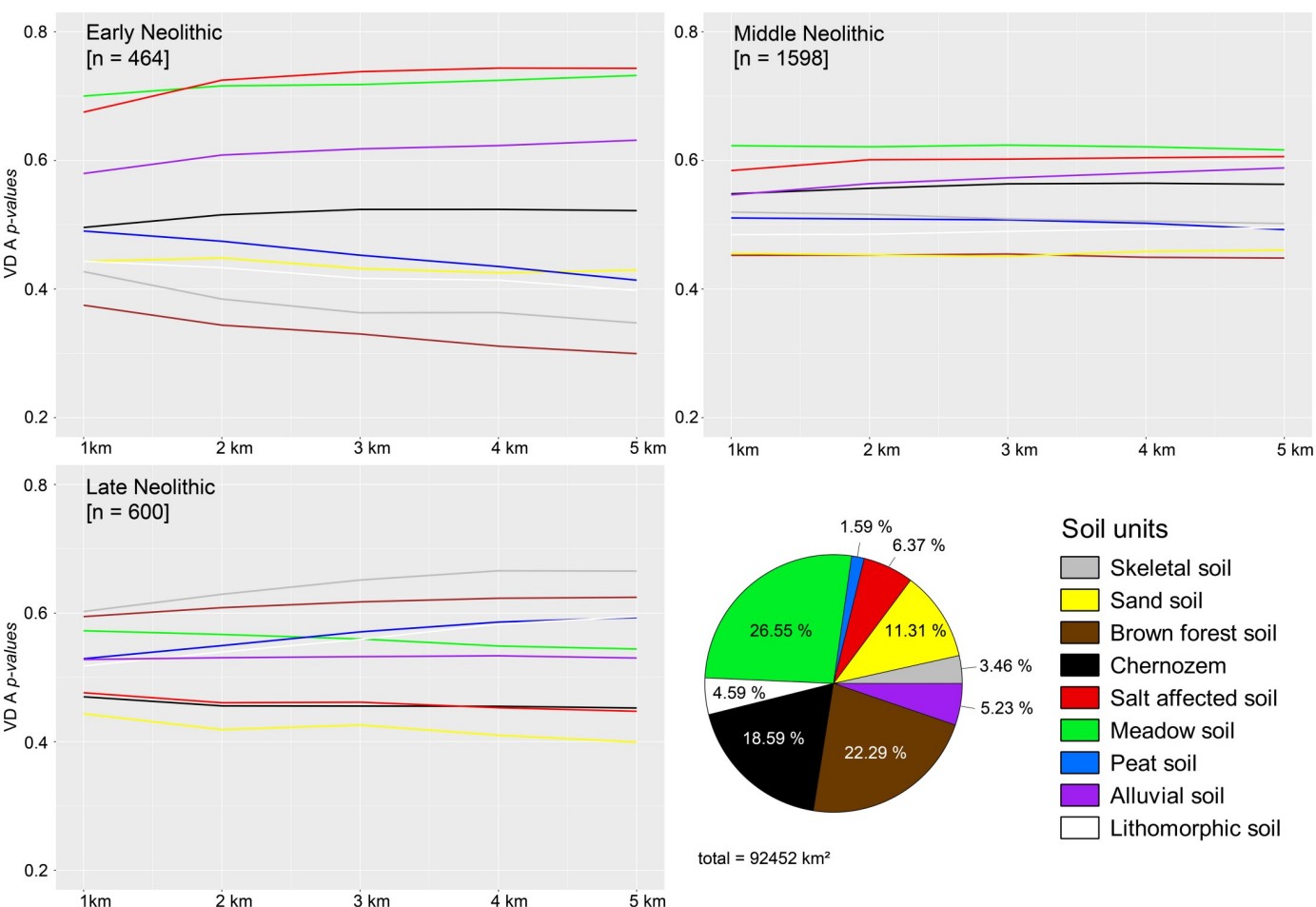

**Fig 6.** *P-values* **of the VD A statistics of early (n = 464), middle (n = 600), and late Neolithic (n = 1598) sites in Hungary indicating the geographical and chronological control of soil preferences and avoidance.** Early Neolithic 'cultural groups' significantly preferred meadow, alluvial, and salt affected soils and significantly avoided brown forest soils, sandy and stony soils. Chernozem soils did not play a major role in site location preferences. The Middle Neolithic site distribution shows similar results but less significant. The Late Neolithic shows increasing impact of brown forest soils and stony and skeletal soils. Loess-covered areas seem to have played a minor role during the entire Neolithic (see **S1**–**S3 Tables**).

against a regional random comparison dataset shows that saline soils can be considered a major site preference during the EN (**Fig 8B**). The data further underlines the preferences of hydromorphic soils within close distance to the sites and the overall avoidance of sandy soils and Chernozem soils on top of the uneroded palaeolevees, which were characterized by low groundwater levels and increased water permeability. In combination with general low annual precipitation totals in the Carpathian Basin, agricultural development would have been tight to the lower parts of the floodplain where direct access to fresh water played a decisive role in the formation of early dwellings or stationary camps. Particularly striking is the non-preference of Chernozem soils during the EN, with a special emphasize on the Körös comparison sample. The maximum intensity of sites at very low values highlights the considerable distance between the site location and the nearest available Chernozem patches (**Fig 8C**).

## Soil preferences during the middle Neolithic

During the MN, there is a clear signal of geographic relocation processes and a spread from the EN core regions along the river Tisza to Transdanubia in the west and the northern

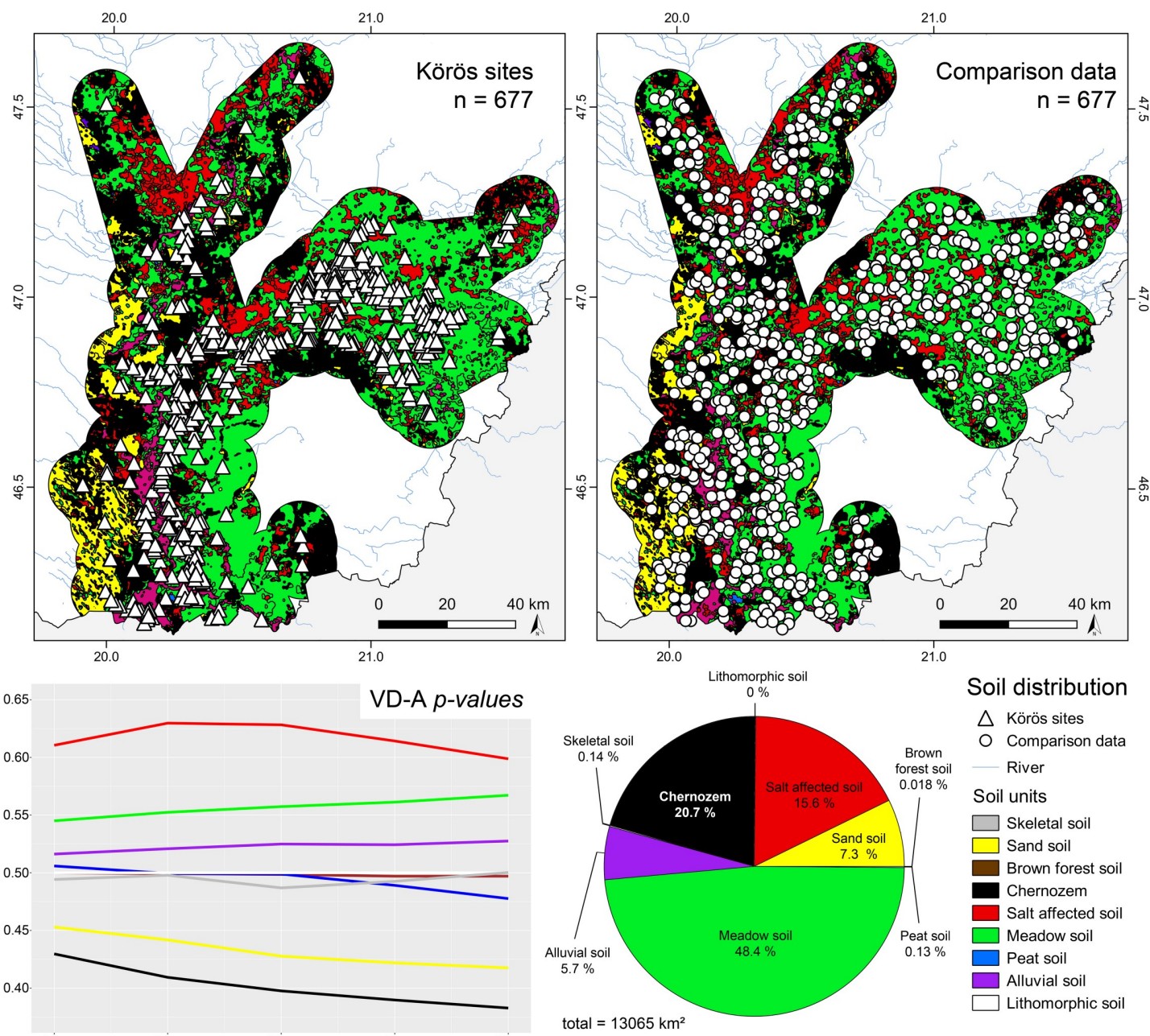

**Fig 7. Distribution of the Körös 'culture' sites (n = 677), the random comparison data sample (n = 677), and the soil units in the study area.** The *p-values* of the VD-A statistics show the significant preferences of salt affected soils, meadow soils, and alluvial soils and further highlight the strongly significant avoidance of Chernozem soils and sand soils. The other soil units did not show significant preferences or avoidance of the Körös 'culture' site occupation.

Hungarian range. This 'cultural spread' should be characterized by the occupation of different soil types, which are increasingly formed by lithic soils. However, from the VD-A statistics, no significant preferences for these soils can be observed in the sample and hydromorphic soils dominate the location factors. Also, the *rhohat* plots do not reveal any significant impact of lithic soils during the MN. There is a little increase in site intensity on skeletal soil, but this can also be explained by the increasing spatial heterogeneity and the larger dataset of sites. The secondary minor peak of sites with high values for brown forest soils can be explained by the

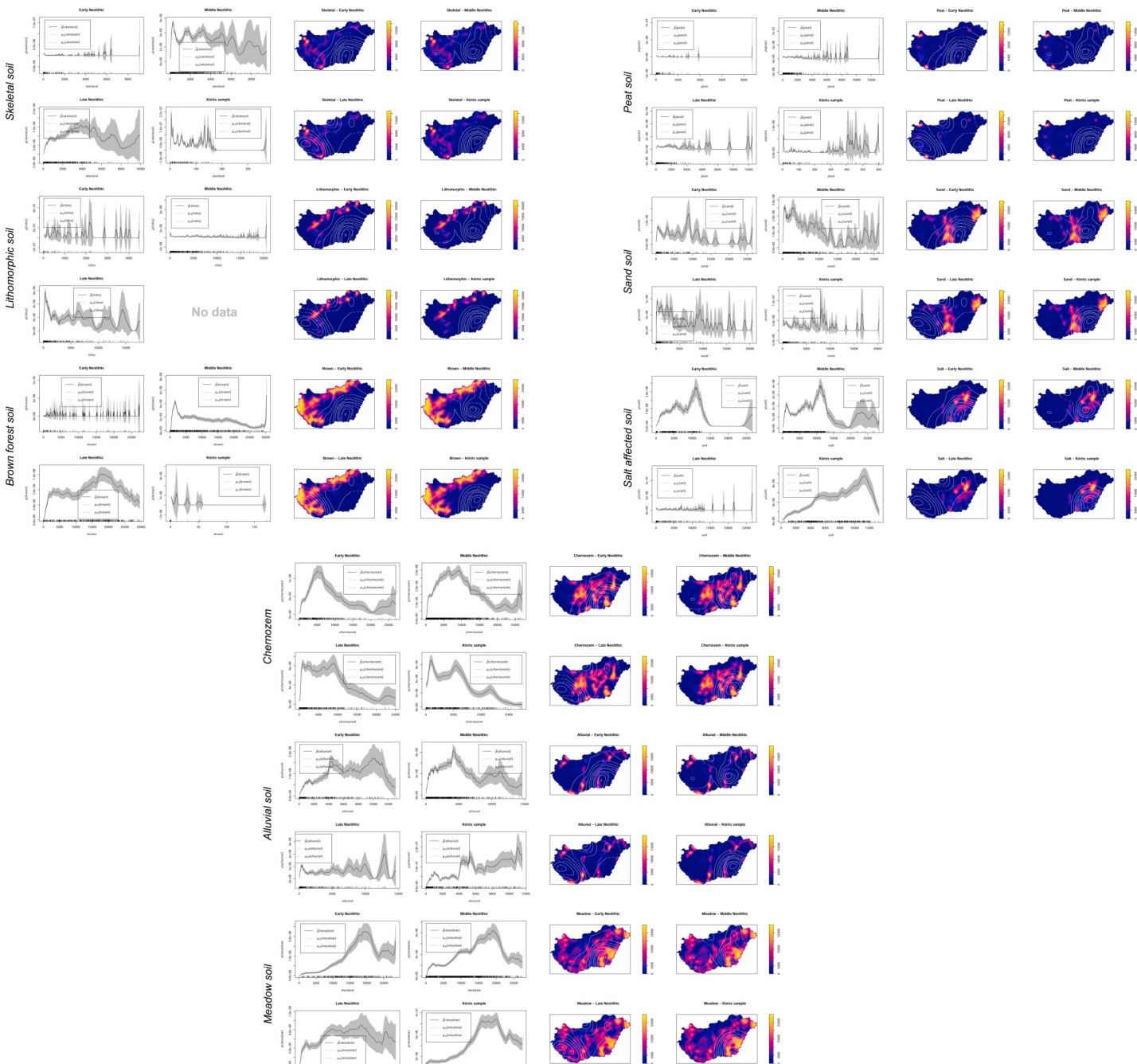

**Fig 8. a.** *Rhohat* plots of the point intensity as a function of the soil explanatory covariate (skeletal, lithomorphic, and brown forest soil). Each soil focal raster was plotted with the kernel density estimation contours to facilitate the interpretation of the plots. **b.** *Rhohat* plots of the point intensity as a function of the soil explanatory covariate (peat, sand, and salt affected soil). Each soil focal raster was plotted with the kernel density estimation contours to facilitate the interpretation of the plots. **c.** *Rhohat* plots of the point intensity as a function of the soil explanatory covariate (Chernozem, alluvial, and meadow soil). Each soil focal raster was plotted with the kernel density estimation contours to facilitate the interpretation of the plots.

above-mentioned expansion of sites towards the northern mountain ranges, which shows that the site preferences are controlled by both, the environmental covariate, and the gradual movement of sites towards the north and the west. The plots further reveal that there is a significant

non-preference of sites for sand-covered soils of the DTI but a continuously very high preference for saline soils in the catchment (10 km). This can be linked to continuous occupation of the river Tisza floodplain region. During the MN, the VD-A p values show medium strong correlations with preferences for meadow and saline soils but the *rhohat* plots underline the continuous importance of hydromorphic soils in the catchments (**Fig 8C**). Chernozem soils became more important but did not reach a significant level. The other soil units did not play a major role in decision-making of location parameters.

## Soil preferences during the late Neolithic

The shift to the LN is more significant as indicated by the site distribution map and the KDE (**Figs 4 and 5**). Large parts of Transdanubia were occupied by the LN 'cultural groups' with concentrations around Lake Balaton. These shifts are reflected in the preferred soil types (VD-A), which now show significant correlations of site occupation and lithomorphic and skeletal soils as well as brown forest soils. The *rhohat* intensity plots confirm the preference for these soil types with maximum values for brown forest soils. The KDE's in Fig 7A point towards a strong shift of site distribution towards the northern and western margins of the Carpathian Basin, which are partly covered with brown forest soils. Consequently, the increasing site intensity as a function of the covariate can also be explained by cultural translocation. Sandy soils and Chernozem soils were still significantly avoided by LN agricultural groups. Saline soils show significant non-preference, which is most likely caused by the limited geographical extent of this soil type in the western part of Hungary and the above-mentioned shift in site distribution.

## Geological and sedimentological site preferences

Soil development is strongly related to geological and sedimentological conditions, and eventually to topographic location parameters of the site's catchments. Preference or avoidance of particular soil patches can further be influenced by the regional geomorphological settings in Hungary, which developed during the Pleistocene and the Early Holocene. Consequently, the pedological analyses were cross-checked with the geological and topographic signal to estimate the geographical bias of the data (**Figs 9 and 10**). From the geological data, a strong preference for Quaternary fluvial deposits can be observed during the EN compared to a random point sample in the VD-A analyses. This signal is confirmed by the Körös sample, however with less significance and a stronger geographical dependency by the EN sample concentration in the eastern part of Hungary, which shows topographical homogeneity through extensive wetlands and remnants of Holocene oxbow lakes and periodically flooded palaeochannels. The minor significance of the Körös sample is caused by the random point distribution within the Körös sample mask, which provides random points that are almost equally distributed over the homogenously dispersed fluvial sediments. The other geological units played a minor role in the Körös site location preferences, mostly because of absence in the data. In the entire EN sample, however, sand deposits and silt and loess-covered areas were significantly avoided by the 'cultural groups'. That confirms the results from the soil distribution analysis, which would rule out the importance of Chernozem soil units during the EN agricultural development. However, these results were most likely generated by the generalized and harmonized geological data in the study area.

During the MN, the avoidance of sandy areas and the preference of humid alluvial floodplains remained significantly high even though strong spatial site relocation can be observed in the western and northern part of Hungary. During the LN, a major shift in site location parameters occurred, which is visible in the geological and the pedological data. Loess deposits and surprisingly mudstones experienced stronger occupation while most of the other units did

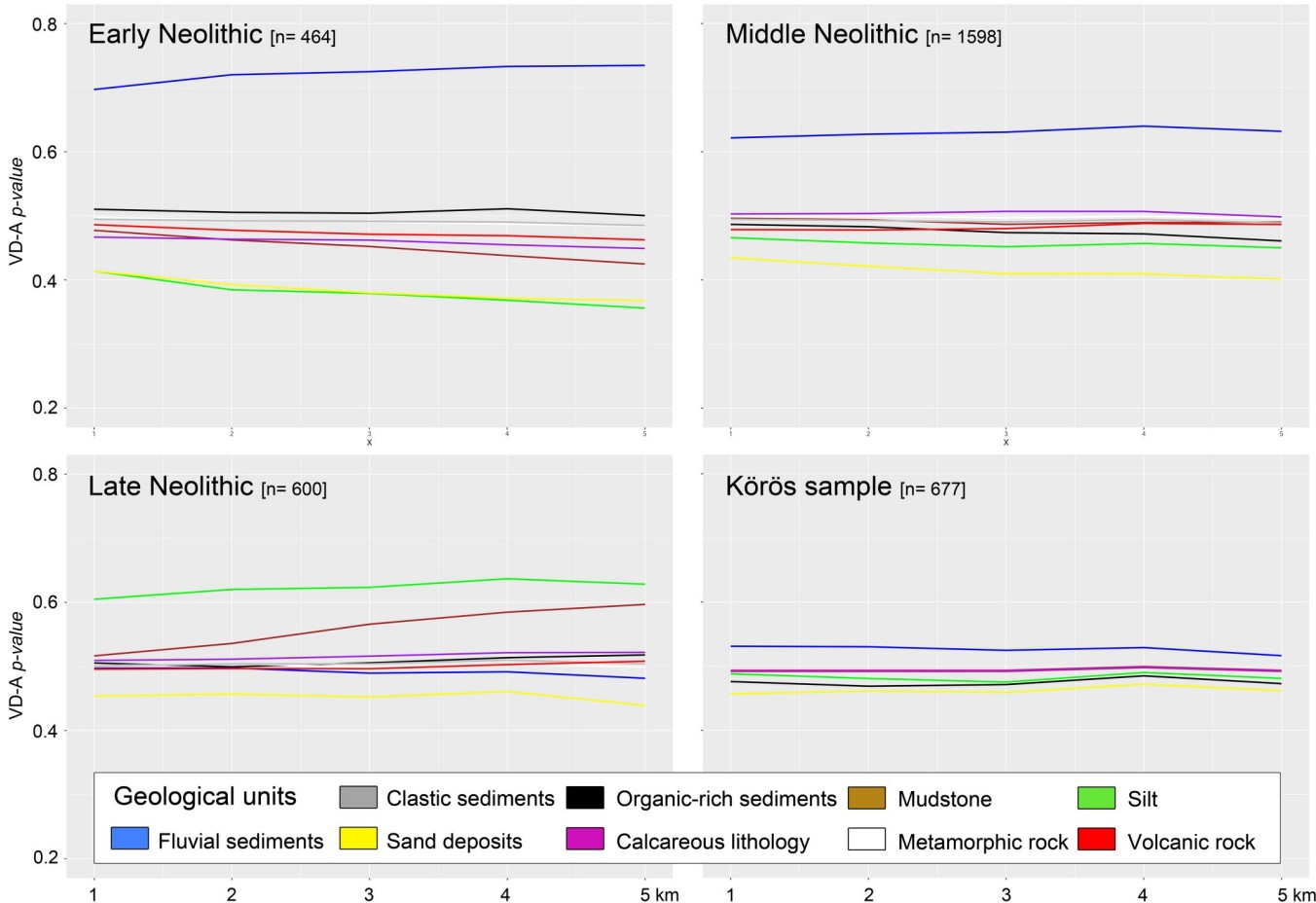

**Fig 9. VD-A statistics of the geological units and the EN, MN, LN, and the Körös sample.** There is a significant preference for alluvial sediments throughout the EN and the MN. The LN sites show different signals with a clear preference of silt deposits (loess). Soil that developed over sand deposits were avoided by all 'cultural groups' during the Neolithic period.

not play a significant role in the choice of a suitable settlement spot. Sandy soils were continuously avoided by LN groups.

## Hydrological and topographical site preferences

The geographical shift from the eastern floodplains to Transdanubia in the west is characterized by the innovative occupation of altered topographic variables during the transition from the Middle to the Late Neolithic period. As seen from the homogeneous Quaternary deposits, the preferences for rather flat and low-lying activity ranges prevailed throughout the EN, which is also confirmed by the Körös sample VD-A statistics and the *rhohat* plots of the elevation and slope (**Figs 11 and 12**). The plots clearly show a significant site intensity on low-elevated and non-sloping areas throughout the EN. Interestingly, there is already a considerable change in topographic elements in the catchment composition during the transition from the EN to the MN period when higher slope gradients were significantly preferred. This can also be due to the general shift towards the northern hilly margins of the basin–as indicated by the KDE plotted on the slope gradient and the elevation model. Even more surprising is the subsequent transition to the west during the LN period. Instead of occupying the elevated areas in the hilly terrain, like there is a considerable preference of lower-elevated areas and valleys in the mountainous landscapes

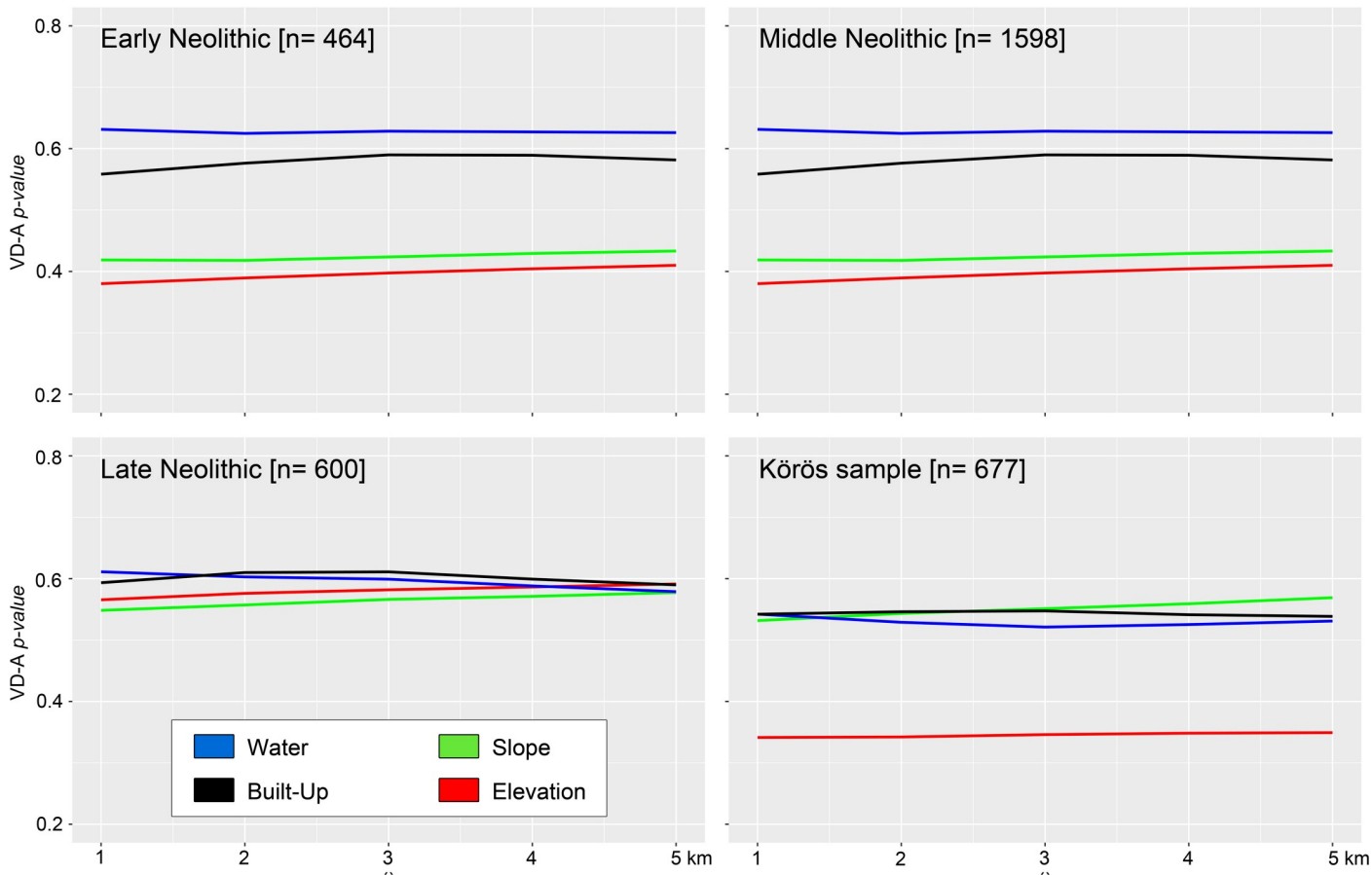

**Fig 10. Additional environmental and modern explanatory covariate.** VD-A p-values of the hydrologic composite (water), slope and elevation as well as the modern built-up within 1 to 5 km distance around EN, MN, LN sites, and the Körös comparison sample.

followed by a secondary peak on higher elevated areas. However, these patches show significant proportions of sloping terrain. The location parameters are confirmed by the attraction through water, which is equally high throughout the Neolithic period. Although one would expect the overall catchment composition of the EN and the MN to be dominated by close access to freshwater, the LN period shows maximum values of sites as a function of nearest water (at a 5 km range). Large distance to water was avoided through the entire Neolithic and obviously no maximum values can be observed, which would represent a site location in water.

From the distance matrix of sites and hydrologic composite, the close distance to water was equally confirmed throughout the entire Neolithic period with up to 18% of all sites located directly within the hydrologic composite area and the general close distance to the nearest freshwater by the rest of the sample. From the VD-A statistics, it can be derived that not much changed during the MN until the LN, where a significant shift from previously avoided slopes and elevated areas to a slight preference of steeper sloping and higher areas prevailed. These results are in accordance with the occupation of large parts of Transdanubia during the LN period by the so-called Lengyel 'culture' [110, 111] and a strong 'cultural' input from the southern Balkanic regions, which shaped the formation of the so-called Balaton-Lasinja 'culture' [112]. The data further highlights the importance of freshwater access, which in turn strengthens the assumption of an increasing occupation of valleys and depressions in Transdanubia during the 'cultural spread' across the western part of the Carpathian Basin.

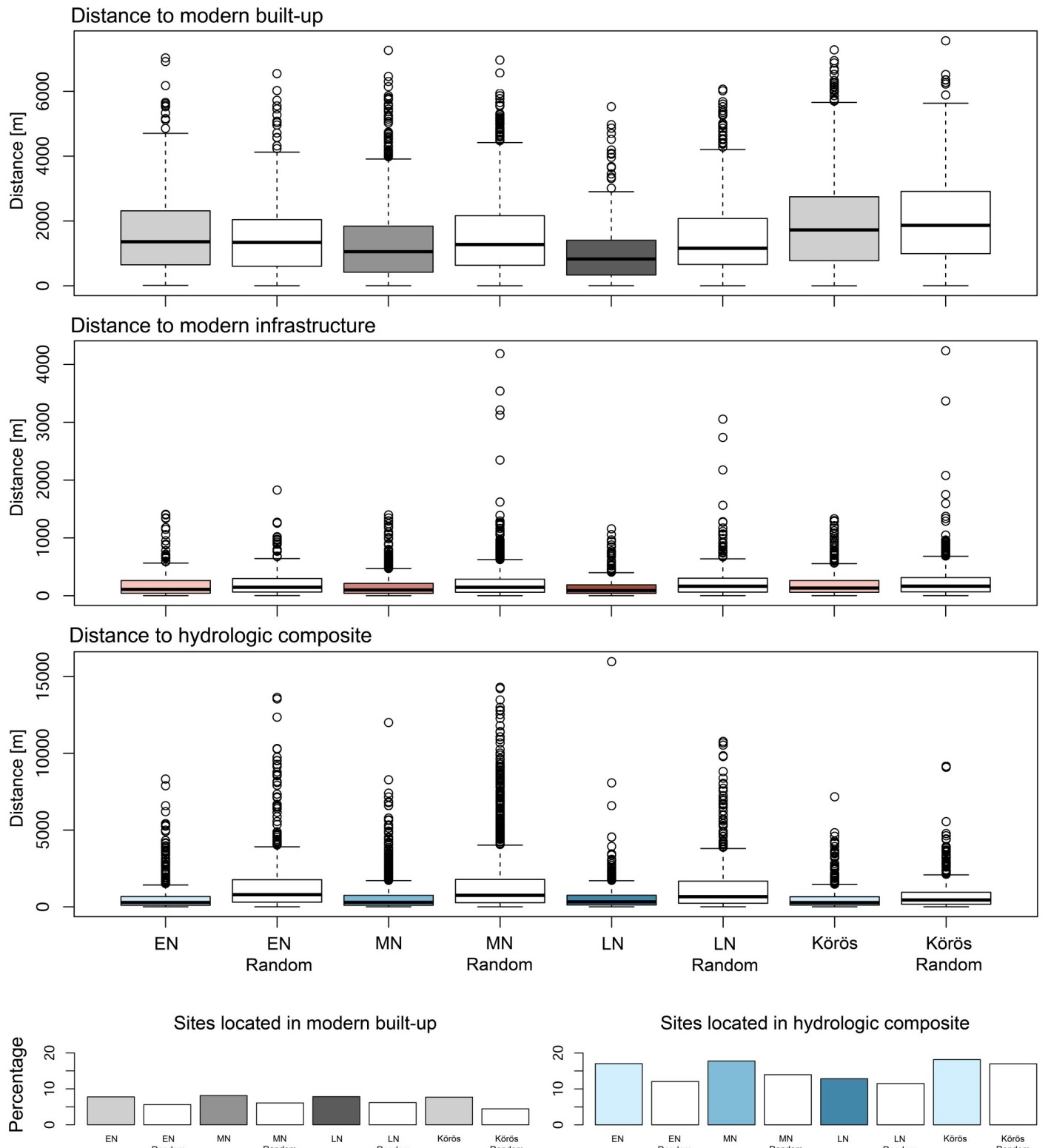

**Fig 11. Distance boxplots of all EN, MN, LN, and the Körös sample sites and their random comparison data to the modern built-up, infrastructure, and the hydrologic composite dataset.** The distance matrix includes all sites that are located outside the respective areas. The barplots show the percentage of sites that fall into the respective extent of the area and were excluded from the distance analyses.

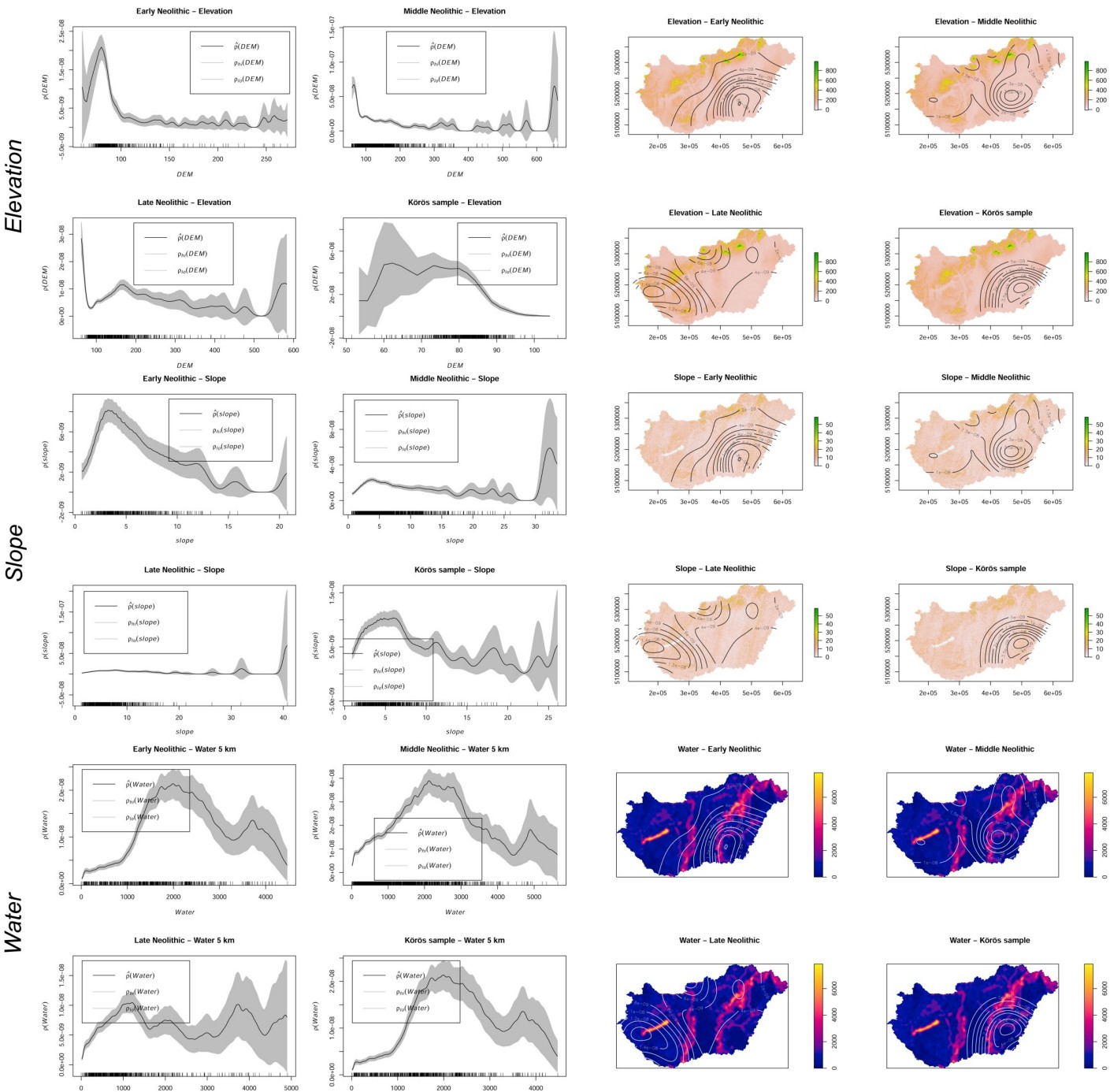

**Fig 12. *Rhohat* plots of the point intensity as a function of the environmental explanatory covariate (elevation, slope, and water).** Each environmental focal raster was plotted with the kernel density estimation contours to facilitate the interpretation of the plots.

## Data bias and restrictions

Empirical data can be biased by a broad variety of impacting factors and particularly archaeological site distribution data can be heavily influenced by modern land-use and the density of the archaeological research activity, including rescue excavations [75, 76, 78, 113, 114]. It is

therefore necessary to cross-check the patterns produced by archaeological site dispersal with the modern extent of rural and urban agglomerations, residential areas, and infrastructural networks as well as the actual reasons behind excavation activity. As discussed earlier, the site dispersal in Hungary is generally impacted by an increasing number of rescue excavations and the simultaneously decreasing time to plan, prepare, and conduct the digs [69, 73]. This has led to an overall mismatch between archaeological departments and institutions and investors and 'land developers' across (and not only) Hungary. From this point of view, the entire country has experienced massive changes in landcover, which favored the discovery of many archaeological features but also created–at least partly–an artificial archaeological distribution. However, some regions in Hungary provide a denser site distribution than others (see also **Fig 5**), which could be caused by more intense research, building activity, or an actually denser archaeological record.

These considerations can be traced through statistical analyses. The VD-A statistics from the modern Hungarian residential and industrial areas reveal a slightly significant impact in the catchment composition of all Neolithic sites (**Fig 10**). The Körös sample did not show significant results, which points towards less inhabited landscapes in the dissemination area of the 'cultural group'. Distance plots between the archaeological sites and the nearest modern built-up and infrastructural units have been produced from the vector data (**Fig 11**). The boxplots confirm the relationship between residential areas and the distribution of archaeological sites in Hungary. From all sites, up to approximately 10% were directly affected by built-up areas. Due to massive landcover change during the 20th century AD, large parts of the country experienced an increase in built-up, particularly in urban agglomerations [115, 116]. These changes can nowadays be traced through satellite imagery, OSM data, and comparison datasets from historical maps [15, 26, 116]. Furthermore, the distance plot to the next available infrastructural entity revealed a very strong correlation between the archaeological site distribution and the road network in Hungary. The random comparison sample shows similar spatial patterns, however with minor significance. Three things can be derived from the matrix: first, there is a very detailed and extensive infrastructural network in Hungary, which affects both the archaeological findability but also any other potential variable. Second, the high number of archaeological sites in close distance to roads mirrors the increased building activity and the high probability to come across archaeological traces. The latter highlights the importance of the Carpathian Basin in terms of archaeological research and the development towards modern society. Finally, the archaeological heritage in Hungary is highly sensitive to massive land-use transformation and particularly built-up and infrastructural change, which poses a great risk of destruction in the course of new construction sites.

To visualize and highlight the spatial variability of all Neolithic sites in Hungary, the whole dataset was tested for the impact by built-up activity and rescue excavations using a comparison artificial surface (**Fig 13**). The artificial mask integrates a slightly larger threshold, which allows an infrastructure buffer of 100 m and consequently contains a higher number of sites directly affected by large-scale construction sites (n = 493). However, it also shows that there are certain clusters, which show significant distance to current building activity and hence could be a function of intense research activity or the above-mentioned actual activity sphere of the respective 'cultural' group–e.g., the Early Neolithic in the lower Tisza region [117, 118]. Furthermore, 151 sites are located in a distance less than 50 m to the next built-up/infrastructure. These sites cluster in the river Tisza floodplain, Transdanubia, Budapest region and the northern part of the Carpathian Basin. However, the cluster align with the clusters of all Neolithic periods and thus mirror the distribution of the total sample and not necessarily a specific site bias (see **Fig 5**).

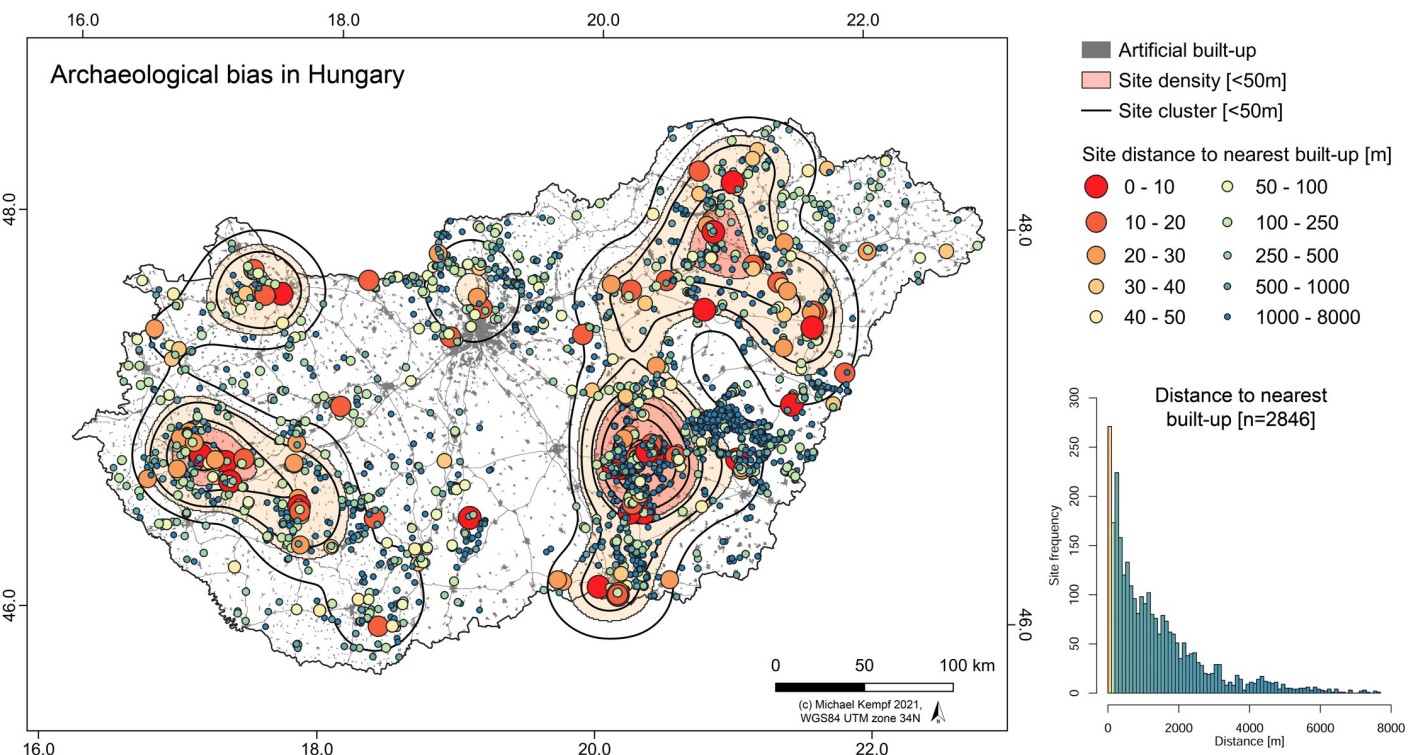

**Fig 13. Archaeological bias in Hungary.** Spatial relationships between sites located outside of the current built-up and infrastructure mask (n = 2846) and the distance to the nearest artificial surfaces. Density estimates of the sites located within a 50 m distance to the nearest built-up are given in red color and contours. Close distance is highlighted by large size and red color, far distance in small circles and blue color. The histogram shows the distribution of the distance values.

## Discussion

The Carpathian Basin certainly represents a region of major importance on the way to human sedentary life and food production. During the Neolithic period, the basin can be divided into two subregions, which developed particular traditions and different 'material cultures' but shared roots in the northern Balkans [8, 119]. The eastern Tisza floodplain region and parts of the eastern bank of the river Danube were the heartland of the Körös 'cultural complexes', which occupied large parts of the Carpathian Basin after 6000 cal BP, crossing the river Maros and the branches of the river Körös to the north [120, 121]. The Starčevo group entered the basin from the south, crossing the river Danube from Serbia and occupied the alluvial wetlands of the Danubian floodplain in Transdanubia [111, 122, 123]. As can be seen from the spatial analyses and the quantitative statistics, the EN occupation was mostly determined by hydro-morphic soil compositions with a considerable number of saline soil patches in close distance to the sites. The overall avoidance of dry, steeping ridges with loess coverage and sand soil units underlines the preferences for low-lying wetlands and floodplains during the EN and the MN period. The palaeolevees, which consisted of Pleistocene and early Holocene sandy deposits often show free-draining soils due to their sedimentological composition. In combination with a low-lying groundwater table, agricultural crop cultivation would be highly vulnerable to periodical drought periods and crop failure [124–127].

In this context, the influence of loess soils in the spread of agricultural development, subsistence food production, and the establishment of permanent settlements has recently gained in importance [20, 21, 23, 26, 27, 128–130]. Results from soil chemical analysis point towards the impact of pyrogenic carbon input from vegetation burning, which triggered Chernozem soil

development after intensified land-use and periodical clearance [29, 30, 131–133]. This is also in accordance with recent discourse about vegetation transition and recovery, the impact of herbivores on natural vegetation, and the maintenance of broad open landscapes through wildfire and the anthropogenic overprint [23, 134–138]. It seems reasonable to not assume extensive Chernozem soil patches and consider site location preferences along the hydromorphic meadow and alluvial soils of the wetlands and the floodplains of the meandering river systems–at least during the EN. This system constantly but not rapidly changed with the establishment of the subsequent Transdanubian Linearbandkeramik (TLBK), which marked the genuine transition to food production in the region [8, 16, 139]. In the counterpart, the Alföld Linearbandkeramik (ALBK) emerged from the Late Körös 'cultural complex' and together they formed the MN transition in the Carpathian Basin, which was influenced by regional 'cultural subgroups' that merged with the later ALBK [140, 141]. Although large parts of Transdanubia were now occupied by the LBK, a continuous preference for wetlands can be observed from the data. Eventually, the LN in the Tisza region was characterized by the development of tell settlements and a more network-like structure of adjacent settlements and local subgroups [142]. Similar processes were apparent in Transdanubia LN, where multifold impulses from the south entered the local TLBK and shaped the later phase of the Neolithic period in the western part of the Carpathian Basin [139, 143, 144]. Finally, the large Lengyel cultural complex emerged, which spread across Europe during the 5th millennium cal BC [110, 145]. During the late phase of the LN Lengyel 'culture', new impacts from the south entered the 'material culture' and particularly the southern part of Transdanubia. The Balaton-Lasinja 'cultural complex' merged Balkanic traditions with late Lengyel features and shaped the transition to the Chalcolithic period. The LN period and the socio-cultural transformation to the early Chalcolithic period is visible in settlement and cemetery structural changes [112, 146] and also in site location preferences, which transformed the landscape and the soil composition. Chernozem soils in Hungary were thought to develop under steppe vegetation during the Holocene, but the manifold types and derivatives are frequently intermixing, and eventually a broad variety of subgroups emerged that include hydromorphic Chernozem variants [24, 52, 147, 148]. Neolithic land-use could initiate the transformation of these soils into modern Chernozems through the intensification of carbon input from increased burning. The spatial spread throughout the MN and the LN period towards Transdanubia and the tendency to occupy more silty soil patches would further enhance the potential of large-scale surface transformation considering the rather long time period of the entire Neolithic.

A potential limitation of this study needs to be addressed, which is a frequently arising problem in archaeological research that aims at understanding large-scale site development and transformation in the context of environmental parameters. Particularly extensive spatial analyses are dependent on modern environmental datasets, such as soil databases, climate interpolations, and the current hydrologic system. Just like the bias by modern land-use and built-up change [78, 113], which impacts the distribution of the current stage of archaeological knowledge, modern datasets cannot be used to entirely 'reconstruct' prehistoric landcover and the archaeological landscapes are not only affected by the physical structure of the landscape but also by perceptional and cognitive patterns [75, 149–154]. Particularly when considering single environmental variables [86, 155], environmental determinism can lead to ignore socio-cultural components in the decision-making processes during human-environment interactions [154]. On the other hand, detailed modern maps or datasets are a compilation of manifold environmental components such as topography and topographical indices, climate, morphometrics, wetness index, channel networks, and many other [63], which provide very detailed information about the ecosystem connectivity [75, 153]. Although the back projection to prehistorical conditions can only be modelled on the basis of extensive coring samples and

palaeoenvironmental proxies, which are cost-extensive and not ubiquitously available, the high number of freely available and high-resolution environmental datasets represents a promising source for current research in landscape and environmental archaeology.

## Conclusion

Digital quantitative analyses in combination with GIS-based data management allowed modelling of large datasets and the evaluation of site location preferences during the Neolithic period in the Carpathian Basin. In this case, over 3000 sites were investigated using environmental modelling and multivariate quantitative statistics to elucidate site catchment compositions at variable distances around Early (EN), Middle (MN), and Late Neolithic (LN) sites. An EN comparison dataset of 677 Körös sites located in eastern Hungary served as reference for the determination of EN land-use and settlement opportunities. From the data, the *Kolmogorov-Smirnov test* (KS-test) and the *Vargha-Delaney A-statistics* (VD-A) were performed, which not only operate at point-based level but integrate environmental catchment compositions from individual neighborhood raster cell analysis. The VD-A offers the potential to reveal preference or avoidance for a specific environmental variable at different radii around archaeological sites. Furthermore, this method was supported by point pattern intensity analyses with variable explanatory covariates using the *spatstat* package and the *rhohat* function in R software. Plots from this method visualize the intensity of points as a function of an underlying control covariate, e.g., environmental datasets. In this study, geological and pedological units, a hydrologic composite based on flood zone and river system, topographical elements such as slope gradient and elevation, and the archaeological bias by modern building activity and infrastructure were included in the analysis. From the chronological differentiation into an EN, MN, and LN period, significant location factors for each period were determined. During the EN, hydromorphic soils over Quaternary fluvial deposits in close distance to water and on flat surfaces were preferred by the first farming 'cultural' groups in the eastern part of the Carpathian Basin. Loess-covered areas were avoided and particularly Chernozem soil did not play a decisive role in settlement location choice. Saline soils, however, played a significant role during the development of early farming techniques–probably due to a shift in dietary habits. However, the flat surfaces with generally low elevation in the river Tisza floodplain, can be considered a typical landscape element in the core area of the Early Neolithic sites. Consequently, the Körös comparison sample has demonstrated that flat zones and freshwater access are more or less ubiquitous and can not be considered decisive parameters in settlement choice.

The MN was characterized by a rapid shift towards the western and northern part of Hungary. Although the database shows almost 1600 sites and a strong geographical shift, no major changes in site preferences occurred during the MN period. This changed significantly during the subsequent LN period, which was characterized by a shift in soil preferences and the occupation of rather stony, lithomorphic soils and an avoidance of Chernozem and sandy soils. Saline soils show decreasing importance as site location parameter. Now, there is also an increased preference for more elevated and slopy regions and the importance of freshwater access remains high.

As discussed earlier, the modern built-up was included as a variable to measure the amount of modern urban and rural agglomerations in the composition of the catchments and to cross-check how much the current archaeological image is actually a function of modern land-use and particularly biased by rescue excavations and a dramatic rise in building activity. All Neolithic sites show significant numbers of built-up over one to five kilometers around each site. Furthermore, there is a strong correlation between archaeological record and the modern infrastructure network, which highlights the bias of the data by modern surface

transformation. In general, modern human landscape development impacts the understanding of the archaeological record by means of modern datasets and an increased findability of archaeological traces in areas with intensive surface transformation and soil movement.

## Supporting information

**S1 Table. Data table.** This table includes the results from the KS-test and the VD-A statistics. The data shows the p-values for the Early, Middle, and Late Neolithic sites as a function of the soil types.
(ODS)

**S2 Table. Data table.** This table includes the results from the KS-test and the VD-A statistics. The data shows the p-values for the Early, Middle, and Late Neolithic sites as a function of geology, topography, hydrologic system, and modern bias by built-up and infrastructure.
(ODS)

**S3 Table. Data table.** This table includes the results from the KS-test and the VD-A statistics. The data shows the p-values for the Körös comparison dataset as a function of all environmental covariates.
(ODS)

## Acknowledgments

I would like to thank Attila Kreiter from the Hungarian National Museum at Budapest who provided the archaeological data that enabled the analysis (the data is available for reproduction upon request from Attila Kreiter and archeodatabase@hnm.hu). The analysis is partly based on soil datasets, which were provided by László Pásztor and his team from the National Academy of Sciences at Budapest. Margaux Depaermentier (University of Basel), Eszter Bánffy (RGK Frankfurt), and Kurt Alt (University of Basel/Danube Private University Krems) added helpful comments to the discussion about Neolithic agricultural development and particularly Eszter Bánffy contributed to the legislation chapter. Martin Hinz (Bern University), Daniel Knitter, Gerrit Günther (both Kiel University), Eli Weaverdyck (Freiburg University), Petr Pajdla, Peter Tóth, František Trampota, Solène Denis, and Jan Kolář (all Masaryk University) were happy to discuss Monte Carlo simulations and the extensive *spatstat* package with me. Anett Osztás (Budapest), Orsolya Láng and Zuszsanna Virág (both Budapesti Történeti Múzeum/Budapest History Museum) provided helpful documents to trace excavation details across Hungary. Finally, this paper benefitted greatly from the constructive suggestions and comments by Piraye Hacıgüzeller and a second anonymous reviewer.

## Author Contributions

**Conceptualization:** Michael Kempf.

**Data curation:** Michael Kempf.

**Formal analysis:** Michael Kempf.

**Investigation:** Michael Kempf.

**Methodology:** Michael Kempf.

**Project administration:** Michael Kempf.

**Resources:** Michael Kempf.

**Software:** Michael Kempf.

**Supervision:** Michael Kempf.

**Validation:** Michael Kempf.

**Visualization:** Michael Kempf.

**Writing – original draft:** Michael Kempf.

**Writing – review & editing:** Michael Kempf.

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
