## [Decision Letter · Decision Letter 0]

23 Jun 2021

PONE-D-21-14576

Take a seed! Revealing Neolithic landscape and agricultural development in the Carpathian Basin through multivariate statistics and environmental modelling

PLOS ONE

Dear Dr. Kempf,

Thank you for submitting your manuscript to PLOS ONE. After careful consideration, we feel that it has merit but does not fully meet PLOS ONE’s publication criteria as it currently stands. Therefore, we invite you to submit a revised version of the manuscript that addresses the points raised during the review process.

In making your revisions, please pay particular attention to the comments of the second reviewer, who makes several recommendations, including to discuss the potential biases due to research history rather than modern land use, to include the equations used for some of the analyses, and to explain the results of hydrological and topographical site location preferences better in the Conclusions. I also concur with this reviewer that the data sharing leaves something to be desired. I understand that you cannot share the raw data with the whole world, and that this is quite common in archaeology, but it would be great to include at least a person of contact that deals with data requests, to at least make it less bureaucratic for others to try to obtain the data.

We look forward to receiving your revised manuscript.

Kind regards,

Radu Iovita

Academic Editor

PLOS ONE

Journal Requirements:

3. We note that Figure(s) in your submission contain [map/satellite] images which may be copyrighted. All PLOS content is published under the Creative Commons Attribution License (CC BY 4.0), which means that the manuscript, images, and Supporting Information files will be freely available online, and any third party is permitted to access, download, copy, distribute, and use these materials in any way, even commercially, with proper attribution. For these reasons, we cannot publish previously copyrighted maps or satellite images created using proprietary data, such as Google software (Google Maps, Street View, and Earth). For more information, see our copyright guidelines: http://journals.plos.org/plosone/s/licenses-and-copyright.

1.    You may seek permission from the original copyright holder of Figure(s) to publish the content specifically under the CC BY 4.0 license.

Additional Editor Comments (if provided):

Reviewers' comments:

Reviewer's Responses to Questions

**Comments to the Author**

1. Is the manuscript technically sound, and do the data support the conclusions?

Reviewer #1: Yes

Reviewer #2: Yes

2. Has the statistical analysis been performed appropriately and rigorously? 

Reviewer #1: I Don't Know

Reviewer #2: Yes

3. Have the authors made all data underlying the findings in their manuscript fully available?

Reviewer #1: Yes

Reviewer #2: No

4. Is the manuscript presented in an intelligible fashion and written in standard English?

Reviewer #1: Yes

Reviewer #2: Yes

5. Review Comments to the Author

Reviewer #1: The present text is an important contribution to the understanding of the neolithization process in the Carpathian Basin. Together with the hydrological conditions, the quality of the soil is one of the most important location factors for early farming societies. Since I am not a specialist in the field of statistical procedures, I would recommend involving another expert with technical expertise. The conclusions from the investigation are definitely very useful and understandable.

Reviewer #2: This article aims to model and analyse the relationship between site locations and a range of environmental variables during the Neolithic period in the Carpathian basin. Specifically, around 3000 sites were taken into consideration with the aim to elucidate on a range of environmental properties of the site catchment compositions during the Early (EN), Middle (MN), and Late Neolithic (LN) periods. The environmental variables that have been taken into consideration as spatial covariates are soil units, elevation, slope gradient, and hydrologic system.

As far as I can judge, the manuscript is well written in standard English. It is also well referenced, but some terminology (e.g. relating to geological units, soil units) may not be accessible to non-specialists. The author uses a range of spatial analysis techniques that are presented in sufficient detail and/or well referenced for expert-to-expert communication, but I believe that at least some of these methodological explanations will not be accessible to non-specialists either. As far as I am aware, the results in the article are original (i.e. not reported elsewhere).

The R code that the author uses for data analysis is published in full detail within the article. As such, the analyses carried out are in principal reproducable. A big BUT here is that data are only “available from the Hungarian National Museum Institutional Data Access / Ethics Committee …. for researchers who meet the criteria for access to confidential data”. This is certainly not an ideal situation in terms of data reuse.

The article forms a unique contribution to the study of the Neolithic period in this area. To my knowledge it is a first in shedding light on the patterns of settlement location and intensity in this region by basing itself on a large number of settlements and detailed environmental and multivariate quantitative statistics. As aimed, it manages to detail changing soil, hydrological and topographic site location preferences during the Neolithic period in the region. It is certainly a welcome contribution to research on human-environmental relations in general and that on the Neolithic of the Carpathian basin more particularly. The title of the article fits the content well.

The article uses a range of point pattern and statistical analysis including the inhomogeneous G-function, Ripley’s inhomogeneous K-function, Kernel density estimates, Kolmogorov-Smirnov test (KS-test) and the Vargha-Delaney A-statistics (VD-A) to assess how much settlements distributions in different periods deviate from Complete Spatial Randomness and how they co-vary with a range of environmental variables listed above. This is thorough methodologically, but I have one major reservation about the overall methodology still regarding the completeness and certainty of the settlement data used. I think it is necessary to clearly indicate (and model) which parts of the region that are being studied here have reliable and complete archaeological settlement data, and data from which parts are limited in terms of certainty and coverage. Archaeology in Hungary is not my area of expertise but it is hard to imagine that there is a very accurate picture when it comes to the location of all the Neolithic sites in the country. I would expect some areas to be better studied than others, but the author does not seem to take this type of bias into account. There is a section in the article that discusses modern land-use and data bias. However, data certainty and coverage issues will not be limited to modern land-use. They will also be heavily influenced by resources available and restrictions in place to carry out archaeological survey and excavations, and data recording which I assume will not be homogenous for the region being studied by the article.

The methodology of the research is generally well-explained and the analytical techniques are performed to a high standard accompanied by high quality images. However, I would like to list the following issues to ensure a clearer presentation of the methodology.

- The formulas in the article are not well-presented in the sense that the article does not explain what coefficients and variables different letters stand for in the formulas (e.g. Ripley’s inhomogeneous K-function on lines 169-171; G-function on lines 183-184). The author refers to the original bibliographical source for these formulas and may be relying on these sources for the full explanation of the formulas. That is fine but it would be useful then to give the exact page numbers for the formulas in the original resources for the readers’ to easily check.

- The explanation of creating continuous soil data on lines 233-241 should be clarified by adding a few more sentences describing the procedure.

- The objectives and process of predictive modelling referred to in between lines 334 and 342 need to be better explained.

Some additional minor remarks:

- On line 64 “tight” should be “tied”.

- It would have been better I think if the “Soil Preferences” section starting on line 399 and ending on line 419 was presented at the beginning of the article in the context of the “Environmental Settings” section (starting on line 96).

- Shifting soil preferences during the Neolithic period are well-explained in the Conclusion section but the results of hydrological and topographical site location preferences are not touched upon. This is an unexpected omission that I believe should be corrected.

6. PLOS authors have the option to publish the peer review history of their article (what does this mean?). If published, this will include your full peer review and any attached files.

Reviewer #1: No

Reviewer #2: **Yes: **Piraye Hacıgüzeller

---

## [Author Response · Author response to Decision Letter 0]

4 Aug 2021

According to the reviewer’s suggestions, I have added a new chapter to the manuscript (and two new figures), which is now considering the recent landcover development in Hungary and integrates also the research/rescue excavation activity in the country. Because Hungarian archaeology is dominated by rescue excavations since the 1990’s – mostly because of the massive built-up change and motorway constructions – the high number of archaeological sites is mostly connected to large-scale digs and less to institutional research projects. Also, and that is a crucial point and functions as a reply to Piraye Hacıgüzeller’s critics, the density of Neolithic sites in the Tisza region is most likely caused by the actual spread of agricultural groups across Hungary and could be considered almost realistic – with all caution of this terminology! During the revision (and throughout the past 2 years), I have further been in close discussion with Eszter Bánffy, who assisted in writing the legislation chapter and who added several personal comments, which enabled this critical part of the manuscript. 

In general, I have taken into account all of Piraye Hacıgüzeller’s suggestions and shifted parts of the soil preferences to the environmental settings, explained the continuous data creation in more detail (adding an example), and cited the original pages of the functions. Also, the number one issue, the data availability, has been solved by an email from Attila Kreiter, who allows to cite his name and email address for further data request:

“Sure, no problem, you can refer to the Hungarian National Museum, and/or me, or instead of me perhaps the email address of the database (archeodatabase@hnm.hu) so whoever handles the database in the feature will receive the request.” (email from Attila Kreiter, 30th of June 2021).

I also added the results from the other covariables to the conclusion section and hope that this has solved the last issue from the review process. Other minor issues can be tracked by the “tracked changes” version of the manuscript in which I also added some comments to reply directly to suggestions by the reviewers.

In general, I very much appreciated the super-constructive comments by Piraye Hacıgüzeller and I am already looking forward to continue working with the dataset and probably re-consider the problem of data bias for the next case study. Also, I liked that she has made public her identity to contribute to a transparent review process!

For now, I am looking forward to the reviewer’s and the editor’s decision and to future comments!

Best regards and again thank you for considering this article for publication,

Brno/Freiburg, 2nd of July 2021

Michael Kempf

---

## [Editor Report · Decision Letter 1]

22 Sep 2021

Take a seed! Revealing Neolithic landscape and agricultural development in the Carpathian Basin through multivariate statistics and environmental modelling

PONE-D-21-14576R1

Dear Dr. Kempf,

We’re pleased to inform you that your manuscript has been judged scientifically suitable for publication and will be formally accepted for publication once it meets all outstanding technical requirements.

Kind regards,

Radu Iovita

Academic Editor

PLOS ONE

Additional Editor Comments (optional):

Thank you for taking the reviewers' questions seriously and implementing their suggestions into the manuscript. One last request is to provide the quantities referred to in the equations on pages 10 and 14 of the manuscript. It is true that those are reviewed in Nakoinz and Knitter's book, but it's not clear what each variable refers to and readers should not have to reach over to that book for understanding the quantities.
---

## [Editor Report · Acceptance letter]

6 Oct 2021

PONE-D-21-14576R1 

Take a seed! Revealing Neolithic landscape and agricultural development in the Carpathian Basin through multivariate statistics and environmental modelling 

Dear Dr. Kempf:

I'm pleased to inform you that your manuscript has been deemed suitable for publication in PLOS ONE. Congratulations! Your manuscript is now with our production department. 

Kind regards, 

on behalf of

Dr. Radu Iovita 

Academic Editor

PLOS ONE